# Quantifying the trade-offs between renewable energy visibility and system costs

Tsamara Tsani [1,2,7] ✉, Tristan Pelser[1,2], Romanos Ioannidis[3,4], Rachel Maier [1,2], Ruihong Chen [5], Stanley Risch [1,2], Felix Kullmann[1], Russell McKenna[5,6], Detlef Stolten[1,2] & Jann Michael Weinand [1,7]

Visual landscape impacts on scenic and populated places are among significant factors affecting local acceptance of large-scale renewable energy projects. Through the combination of large-scale reverse viewshed and techno-economic energy system analyses, we assess their potential impacts for nationwide energy systems. In our case study of Germany, moderate consideration of visual impact by placing renewables out of sight of the most scenic and densely populated areas does not have a significant impact on future energy system costs and design. In contrast, in scenarios assuming high sensitivity to visual impacts, annual energy system costs would increase by up to 38% in 2045. The energy system's resilience would also be compromised due to the increasing reliance on green hydrogen imports and the uncertain mass adoption of rooftop photovoltaics. Our analytical framework facilitates careful planning that considers the visual impact of renewable energy infrastructure, thus enabling socially acceptable deployment while understanding the implications for system costs and transformation pathways.

A key component in mitigating climate change is the substitution of conventional fossil fuels with sustainable, low-carbon, renewable energy sources[1]. Despite their economic competitiveness due to strongly decreasing costs of on- and offshore wind power[2-4] as well as solar energy[5,6], the current growth dynamics of renewable energies are not sufficient to enable 1.5 °C-compatible scenarios[7]. In Europe, and particularly in Germany, after years of record capacity expansion, the growth rates have been declining sharply[8-10]. The decline has been attributed to a number of factors, including the lengthy permitting process[9,11], supply chain issues, and insufficient grid expansion[12]. Above all, local opposition to renewable energy technologies, particularly wind turbines, has been identified as one of the most significant barriers to their deployment[3,8,13-15].

The construction of on- and offshore wind turbines are increasingly opposed by local stakeholders[14,16] with the visual impact of the turbines on the landscape being the main concern[15,17-23]. In particular, turbine installations are rejected in landscapes with high esthetic quality, while they are accepted more easily in less beautiful landscapes[24-29]. Although solar energy generally has less impact on the landscape[30] and causes less public opposition[31,32], the visual impact, especially of large-scale photovoltaics (PV), is seen critically[33], and in specific regions, the opposition is stronger than toward the wind[34]. Together with other externalities such as noise, threats to wildlife, and decline of property prices, the visual impacts of renewable technologies appear to also diminish for local residents with increasing distance from the plant[28,35-37]. It is crucial to address these concerns, as the integration of renewable energy sources has the potential to exert adverse local impacts in social, environmental, or economic terms if not planned with sufficient consideration[38].

[1]Institute of Climate and Energy Systems - Jülich Systems Analysis (ICE-2), Forschungszentrum Jülich GmbH, Jülich 52425, Germany. [2]Chair for Fuel Cells, RWTH Aachen University, Aachen 52062, Germany. [3]Department of Water Resources and Environmental Engineering, School of Civil Engineering, National Technical University of Athens, Heroon Polytechniou 5, Zografou 157 80, Greece. [4]Department of Architecture, Built Environment, and Construction Engineering, Politecnico di Milano, Milano 20133, Italy. [5]Chair of Energy Systems Analysis, ETH Zürich, Zürich 8092, Switzerland. [6]Laboratory for Energy Systems Analysis, Centers for Nuclear Engineering and Sciences & Energy and Environment, PSI, Villigen 5253, Switzerland. [7]These authors contributed equally: Tsamara Tsani, Jann Michael Weinand. ✉e-mail: t.tsani@fz-juelich.de

The primary planning approach to mitigate and assess visual landscape impacts from renewable energy projects is visibility analysis[39,40]. Visibility analysis can be performed in a variety of ways, including visibility maps generated from viewshed analysis, 3D simulations, and photomontages[41,42]. Nonetheless, when planning projects at large spatial scales, for example for regional or national scale, the aforementioned methods cannot be efficiently utilized. The reason for this is in the case of viewshed calculation, the analysis is based on a line-of-sight test[43] that is carried out from the perspective of an examined project. Consequently, the exact locations of all examined projects must have been determined first. This is not possible at large spatial scale, where the locations of potential projects is still under investigation. Therefore, the application of visibility analysis for planning so far is limited to small spatial scales[44–48] or for the purpose of impact assessments[49,50]. The shortcomings of conventional viewshed analysis, however, can be overcome by reversing their setup, i.e., performing the analyses from the perspective of the landscape areas that are to be protected, rather than from the perspective of the examined projects. This reverse viewshed assessment has the potential to be applied to large-scale planning of renewable energy deployment[51], and it will be utilized in the present study.

Given the former limitations of conventional visibility analyses, only a few studies[25,52–54] have attempted to incorporate visual landscape impact considerations into nationwide energy system analyses or transformation studies. Furthermore, the studies have only considered visual landscape impacts in simplified ways, or have focused primarily on visual impacts from onshore wind systems[54]. For instance, one study[52] substituted wind turbines entirely with PV systems, which presumably have lower visual impacts. Other studies[25,53] excluded onshore wind placements in scenic areas.

In this study, we combine mathematical, techno-economic, and landscape planning approaches to address the following research questions: would it be possible to install a national renewable energy system that is not visible from scenic or densely populated areas and would potentially encounter significantly less local opposition? If so, what are the associated costs of such an energy system design? We seek to answer these questions for a case study of Germany by first determining the reverse viewsheds for all populated and scenic places in Germany, from individual persons to wind turbines with 130-m hub height and photovoltaic plants with 2-m height. Subsequently, we determine the techno-economic potential for on- and offshore wind, and open-field PV that is not within the viewsheds. Finally, we compare energy system transformation costs by 2045 with and without renewable energy technologies excluded by viewsheds, utilizing a national energy system optimization model. This way, we integrate large-scale reverse viewsheds of renewable energy infrastructures into feasible potential analyses and techno-economic energy system planning.

## Results

### Reverse-viewshed maps and scenarios

The reverse viewshed analyses were conducted on the entirety of Germany using the Copernicus EU-DEM (Digital Elevation Model) version 1.1 with 25-m grid resolution[55]. The observer points were positioned at every centroid of a 1-km² grid that have underlying metadata of scenicness level and population density. In total, viewshed analyses were conducted from 357,588 viewpoints to determine theoretically visible areas from each viewpoint with different scenicness and population density levels. The setup of the basic parameters of the viewshed analysis are presented in Fig. 1a. The generated reverse viewshed maps were then incorporated as exclusion zones in the land eligibility assessments for renewable energy capacity potential calculation. The base scenario for our land eligibility assessment follows the study of ref. 56 that considers legal, technical, geographical, environmental, and cultural preservation restrictions as listed in

Supplementary Table 1. An example of a small-scale application of incorporating reverse viewshed maps into land eligibility assessment for the district of Aachen is presented in Fig. 1b.

In total, nine different visibility scenarios were considered accounting for people's sensitivity to potential visual impacts of renewable energy technologies in highly perceived landscape importance (i.e., scenic or densely populated areas)[42]. The logic behind the selected scenarios is as follows: first, we determined the areas available for the placement of renewable energy systems that would not be visible from the most scenic places in Germany, i.e., scenicness level 9, at a scale of 1 (low) to 9 (high scenicness)[57]. Subsequently, the thresholds were gradually increased to also exclude renewable energy systems that are visible from lower scenicness levels down to the average scenicness level in Germany (i.e., excluding renewable energy plants visible from scenicness levels ≥ 8, ≥ 7, ≥ 6, and ≥ 5). The last scenario represents our strictest exclusion criteria that reflect the high sensitivity of the population to the visual landscape impacts from renewable infrastructure, even when viewed from the average landscape scenicness. We did not consider lower scenicness levels, as these are mostly in urban areas[8,57], which we incorporated by similarly considering population density levels. For the latter, we started by determining the available areas that are not visible from high-density urban centers (i.e., population density ≥ 5000 people per km²). Lastly, the thresholds were gradually increased to account for scenarios where renewable energy systems are not visible even from areas with lower population densities (i.e., population density ≥ 3500 people per km², ≥ 1500 people per km², ≥ 300 people per km²).

### Renewable energy plants invisible from scenic and densely populated areas

The existing renewable energy plants[58] in Germany are predominantly visible from locations with low scenicness values and low population density (see Supplementary Figs. 1, 2). A total of 65% of onshore wind turbines and 86% of the open-field PV projects are visible from locations with scenicness levels of 6 or lower. As scenicness levels increase, the number of visible plants decreases, until only 3% of the total onshore wind turbines and 2% of open-field PV projects are visible from the most scenic locations (i.e., scenicness level = 9). With regard to population density, this trend is less pronounced for onshore wind, but remains consistent for open-field PV: the majority of existing open-field PV projects (77%) are visible from areas with population density below 1500 people per km². Only 2% are visible from densely populated areas, with a population density above 5000 people per km². For onshore wind, 18% of the total onshore wind turbines are visible from areas with a population density below 1500 people per km², while 12% are visible from densely populated areas.

Germany has a total capacity potential of 398 Gigawatt (GW) for onshore wind, 79 GW for offshore wind, and 669 GW for open-field PV (without visibility restrictions). As the restrictions on visibility based on scenicness become more stringent, the capacity potential of onshore wind and open-field PV are gradually reduced, as shown in Fig. 2. At a scenicness level of 9, the reverse viewsheds exclude renewables that are visible from the most scenic landscapes in Germany, such as the Black Forest and the Bavarian Alps. This results in a reduction of the capacity potential by 10% for onshore wind and 4% for open-field PV. These areas, despite their scenic quality, are inhabited by less than 0.5% of the German population. However, they might be of significant importance as tourist attractions.

Figure 2 shows that the capacity potential starts to decrease significantly as renewable energy plants that are visible from scenicness level 7 or lower are excluded. This trend continues, and leaves only three GW of onshore wind and 44 GW of open-field PV potential in the scenicness ≥ 5 scenario. Designing a renewable energy system that is not visible from areas with the average level of scenicness in Germany would reduce 99.3% of onshore wind and 93.5% of open-field PV

a.

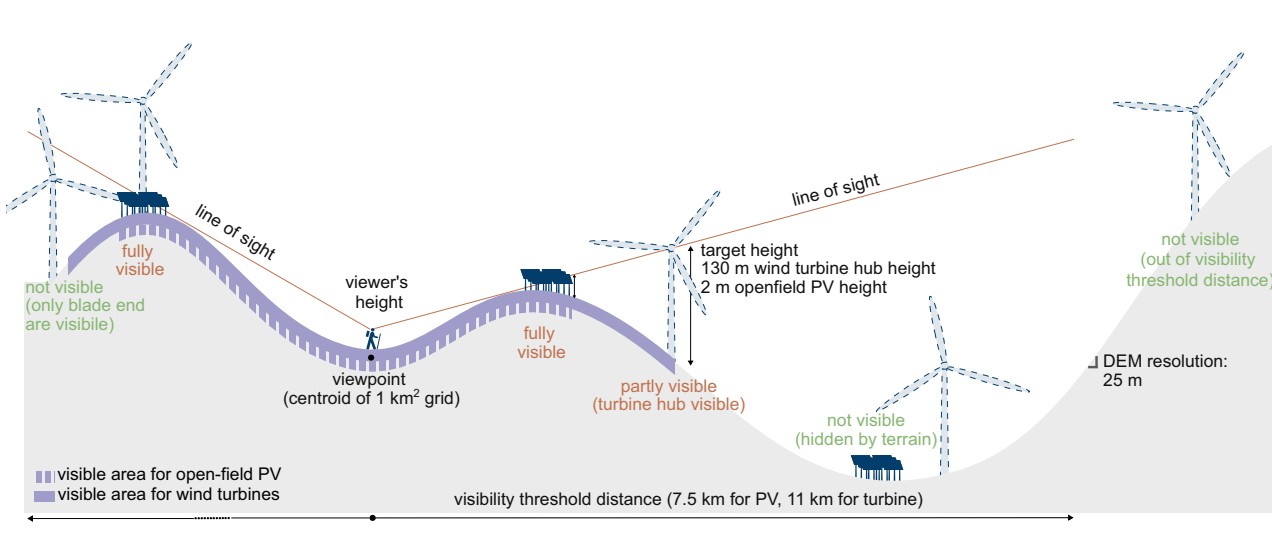

b.

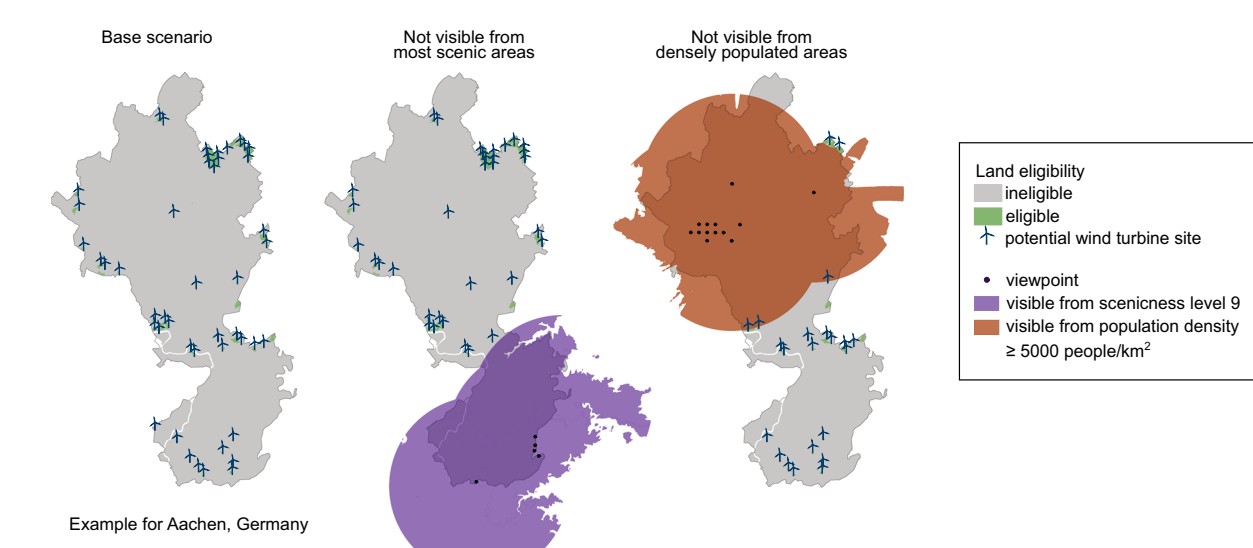

**Fig. 1 | Methodology of reverse viewshed analysis and exemplary use of the generated reverse viewshed maps as exclusion zones in land eligibility assessments. a** Reverse-viewshed analysis is performed from a selected viewpoint to map the areas where wind turbines or open-field photovoltaics (PV), if constructed in the areas, would be visible to people standing at the selected viewpoint. This analysis requires a digital elevation model (DEM) of the area and location of viewpoints that are to be protected from the visual impacts of renewable infrastructures. The EU-DEM v1.1 with 25-m resolution is used for the analysis. The viewpoints utilized are the centroids of a 1-km² grid of Germany with underlying metadata of scenicness and population density level. The visibility threshold distance was set at 11 km for wind turbines and 7.5 km for open-field PV. **b** Example of integrating reverse viewshed maps from viewpoints with high scenicness level (level 9), and high population density (≥ 5000 people per km²) into the land eligibility assessment of onshore wind turbines in the district of Aachen, Germany. The green areas represent the eligible areas for the siting of onshore wind turbines based on legal, geographical, technical, environmental, and additional visibility restrictions.

capacity potential. In contrast, the capacity potentials and eligible areas for offshore wind remain constant across different visibility scenarios due to legal restrictions that require offshore wind turbines to be placed at least 15 km away from the coast[56]. This distance is greater than the assumed visibility threshold of 11 km (see Methods), below which distance a wind turbine has a significant visual impact on the landscape[59]. Furthermore, Fig. 2 also demonstrates that less than 25% of Germany's population resides in areas with above-average scenicness levels. In addition to tourists who frequent these areas, the strictest scenario considered in this study (i.e., scenicness ≥ 5) has the potential to safeguard 20 million residents in these areas from the visual impact of large-scale renewable infrastructure.

Minimizing the visibility of renewable energy infrastructures from high-density urban centers, such as Berlin and other metropolitan areas, would reduce onshore wind potential by 10% and open-field PV potential by 6%. As illustrated in Fig. 2, as the restrictions on visibility based on population density progress, the areas available for renewable deployment continue to decrease. Ultimately, this leaves only 1.5 GW of onshore wind (99.6% reduction) and 68 GW of solar PV (89.8% reduction) potential that is invisible from areas with a population density greater than 300 people per km². In this strict scenario, 87% of Germany's population (~70 million people) would be protected from the visual impact of large-scale renewable energy infrastructure when viewed from their residences. In addition, the results obtained

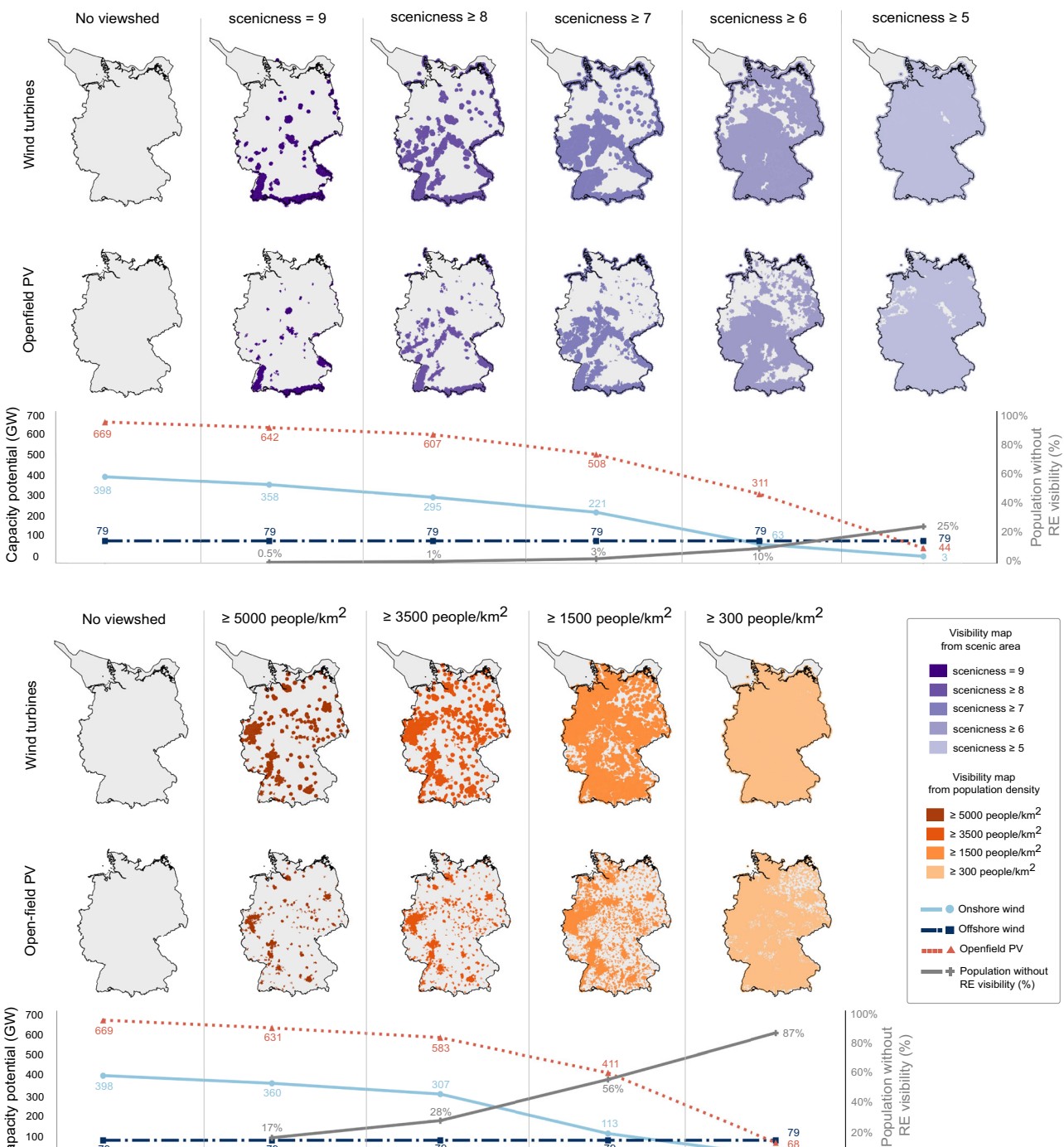

**Fig. 2 | Reverse-viewshed maps and the remaining renewable capacity potential at each visibility scenario.** Reverse-viewshed maps show siting areas for wind turbines and open-field photovoltaics (PV) that are visible from different scenicness and population density thresholds. Each map is utilized as an exclusion zone in the capacity potential calculation, and the remaining renewable energy potential for each scenario is displayed in the subsequent line graphs. The capacity potential, depicted as line graphs, are calculated after taking into account other legal, geographical, technical, environmental, and additional reverse viewshed constraints. The secondary y-axis in the line plots shows the percentage of the population protected from the visual impacts of large-scale renewable infrastructures. Source data are provided as a Source Data file.

across all scenarios depicted in Fig. 2 demonstrate that the visual impact of open-field PV systems on the landscape is less significant in comparison to that of wind turbines.

Furthermore, we conducted a simulation to determine the electricity generation potential and the levelized cost of electricity (LCOE) at each potential site for each visibility scenario (Fig. 3). In the base scenario, 398 GW potential of onshore wind turbines could generate 877 Terawatt hours (TWh) per year, whereas 669 GW open-field PV

systems could supply 728 TWh per year. The graphs again demonstrate that more stringent visibility restrictions result in a reduction of the potential for both onshore wind and open-field PV. It is worth noting from Fig. 3a that gradually excluding wind turbines visible from the most scenic areas in Germany also progressively eliminates turbines with high LCOEs. This indicates that some sites with high visual landscape impacts in Germany coincide with the worst wind conditions, suggesting an alignment of landscape protection with the

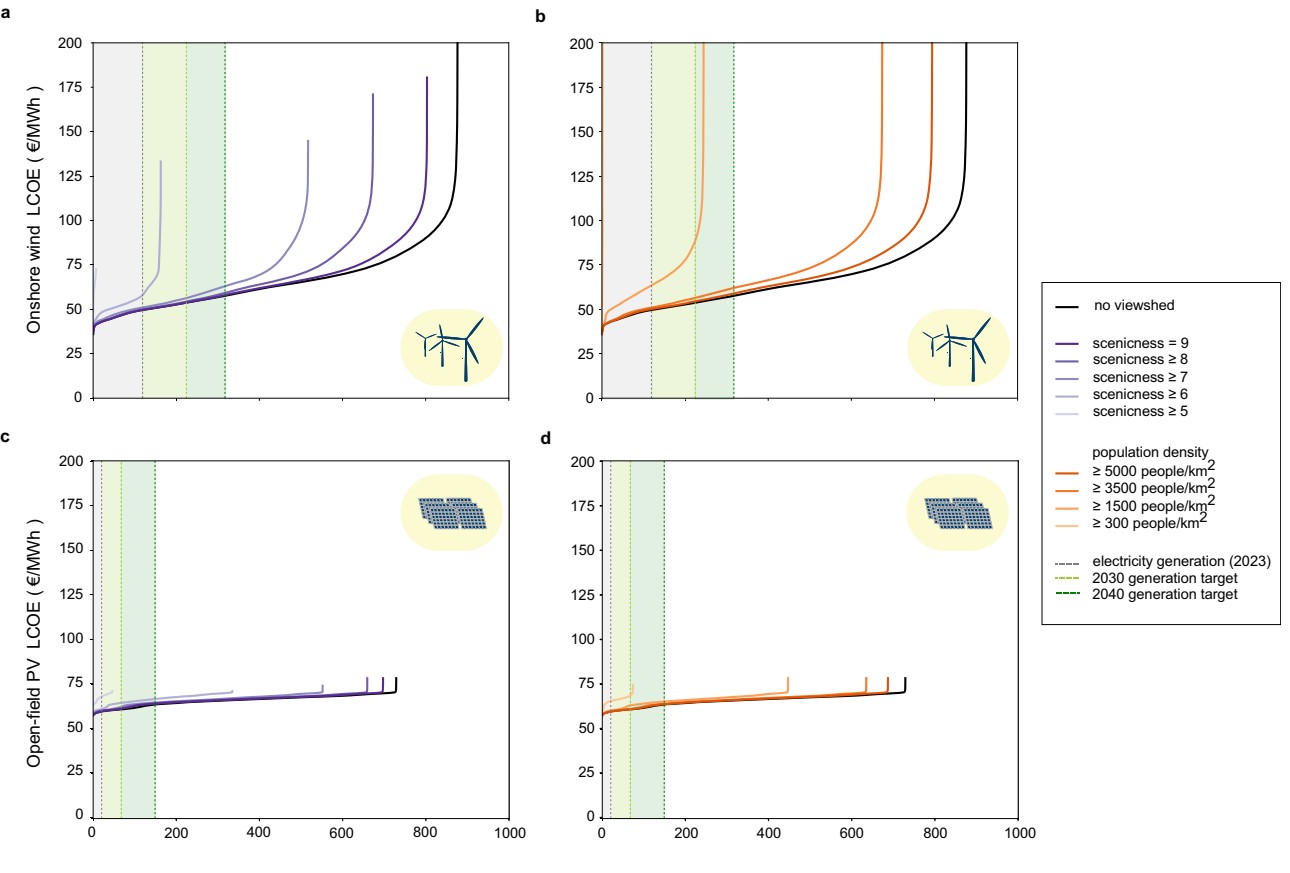

**Fig. 3 | Cost-potential graphs of onshore wind (a,b) and open-field photovoltaics (PV) (c,d) not visible from different scenicness and population density thresholds.** The area in light gray represents annual electricity generation in 2023. The areas in light green and green represent the electricity generation targets according to the German Renewable Energy Sources Act (EEG, 2023) for each technology by 2030 and 2040, respectively. Source data are provided as a Source Data file.

objective of cost-effectiveness. This is not the case, however, for visibility restrictions based on population density. In addition, the LCOEs for onshore wind power vary considerably across locations in Germany, ranging from €36.1 (Euro) per Megawatt hours (MWh) to €471.1 per MWh, with an average of €70.5 per MWh in the base scenario. In contrast, the LCOEs of open-field PV systems exhibit less variation, ranging from €57.9 per MWh to €78.5 per MWh. This suggests that the quality of solar resources is relatively consistent throughout Germany.

In regards to the attainability of political targets, Fig. 3 shows that the electricity generation targets from onshore wind power for 2030 and 2040[60] are not achievable under the two strictest scenarios examined in this study, based on scenicness (i.e., scenicness ≥ 5, ≥ 6) and population density (i.e., population density ≥ 300 people per km², ≥ 1500 people per km²). However, for open-field PV, this is only true for the strictest visibility scenarios considered (i.e., scenicness ≥ 5 and population density ≥ 300 people per km²).

**Impacts on system costs and energy transformation pathways**
Utilizing the remaining renewable energy potential under different visibility scenarios, we identified cost-optimal transformation pathways for the sector-wide German energy system to achieve the greenhouse gas-neutral goal by 2045. We found that excluding large-scale renewable infrastructure that are visible from the most scenic (scenicness = 9) or densely populated areas (population density ≥ 3500 people per km²) does not affect the optimal system cost (see Supplementary Fig. 6). The overall system costs start to gradually increase at scenarios where renewable energy systems visible from scenicness levels ≥ 8 or population density ≥ 1500 people per km² are

excluded. Eventually, restricting renewable energy infrastructures to areas that are not visible from average scenicness onwards (i.e., scenicness levels ≥ 5) results in an increase in the overall system costs by up to €45.5 billion per year in 2045. Furthermore, not employing renewable infrastructures that are visible from areas with an average population density (population density ≥ 300 people per km²) increases system costs by up to €56.4 billion per year in 2045. The majority of these additional costs come from the energy sector, which would experience up to 38% of the cost increases per year in 2045 (equivalent to €23.6 billion per year) under these strictest visibility scenarios. The energy sector considered in this study encompasses electricity generation, imports, and exports.

In scenarios with strict visibility restrictions, the declining electricity supply from onshore wind and open-field PV is substituted by offshore wind and rooftop PV (see Fig. 4c–d). In these scenarios, the massive deployment of distributed rooftop PV necessitates huge storage development. This is reflected in the increases in storage sector costs by up to threefold in scenarios where large-scale wind turbines and open-field PV are not visible from scenicness levels ≥ 5 and population density ≥ 300 people per km², compared to the base scenario (Fig. 4a–b). In these strict scenarios, marginal reductions in the infrastructure sector costs can also be observed, as the distributed use of rooftop PV may reduce the necessity for grid expansions.

In the base scenario, offshore wind and rooftop PV collectively contribute to only 12.6% of the electricity supply in 2045 (Supplementary Fig. 5). However, as the visibility restrictions progress, the supply from open-field PV and onshore wind is gradually replaced by

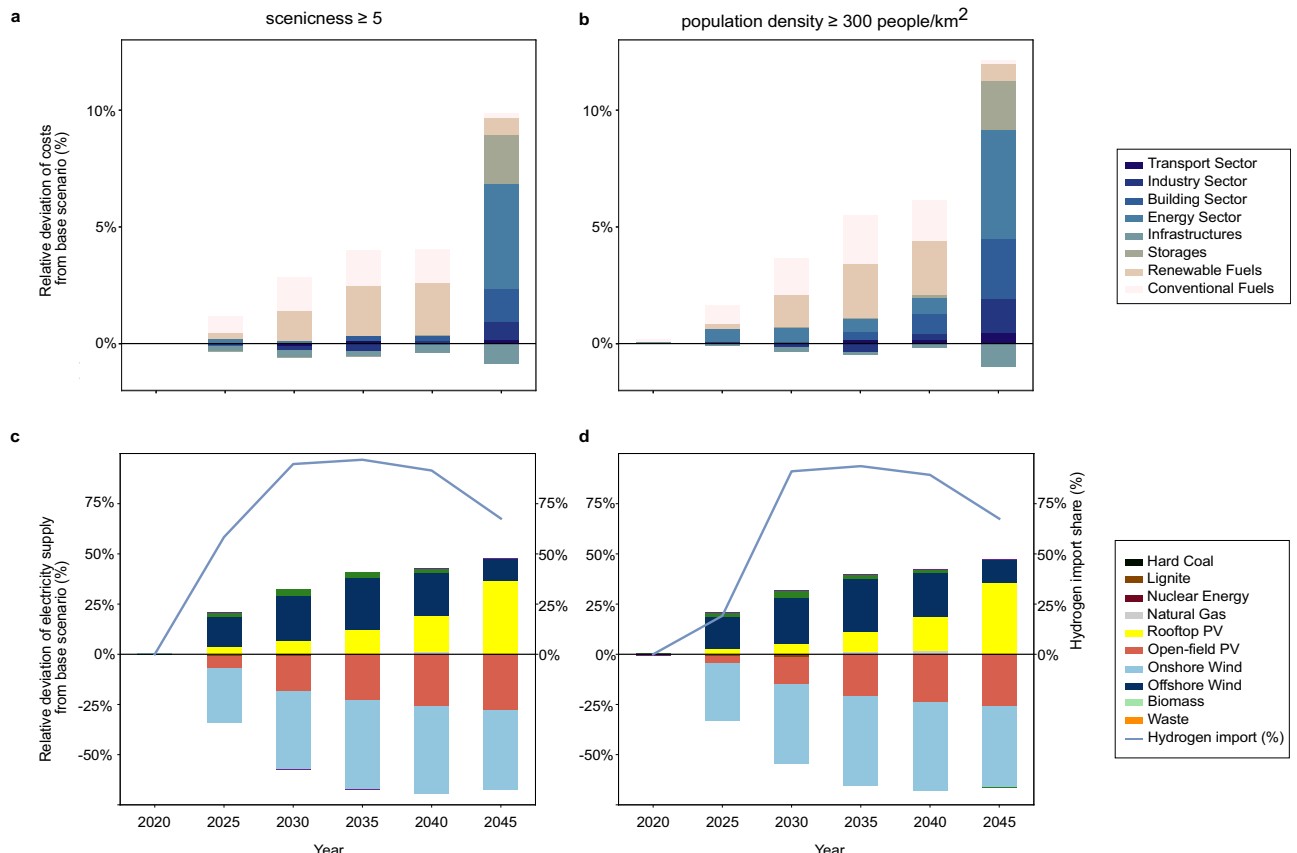

**Fig. 4 | Relative cost deviation by sectors and relative deviation of electricity supply by sources when large-scale renewable energy plants are not visible from scenicness levels ≥ 5 and population density ≥ 300 people per km², compared to the base scenario. a**, **b** The cost deviation is shown for each economic sector and important energy-related sub-sectors. The energy sector accounts for the domestic energy supply. The infrastructure sector accounts for grid costs. Renewable and conventional fuels represent costs for imported fuels. Building and transport sectors are not as strongly affected by the various visibility scenarios as the energy sector. In these two scenarios with the strictest visibility restriction, the cost from the energy sector increases by 38% in 2045 compared to the base scenario (equivalent to €23.6 billion). **c**, **d** Reduction in electricity supplies from onshore wind and open-field photovoltaics (PV) are substituted by rooftop PV and offshore wind. The blue line shows the need to increase hydrogen imports to meet demand in these two scenarios with the strictest visibility restrictions. Source data are provided as a Source Data file.

rooftop PV and offshore wind power (Fig. 4c–d). In the strictest scenarios, rooftop PV and offshore wind power account for up to 73% of electricity supply in 2045. This strategy exhausts the capacity potential for rooftop PV[56] and offshore wind in Germany. In the industrial and residential sectors (Supplementary Figs. 12, 13), there are also notable increases in natural gas use at the strictest visibility scenarios. These strategies also necessitate a sudden phase-out of fossil-based fuels in these sectors within only five years, from 2040 to 2045. Furthermore, the massive non-utilization of onshore wind and open-field PV due to concerns regarding their visual impact also affects the green hydrogen supply. As shown in Fig. 4c–d, in scenarios where onshore wind and open-field PV are not visible from scenicness ≥ 5 and population density ≥ 300 people per km², 91–95% of green hydrogen supply comes from import in 2030. This equates to 77 TWh per year of green hydrogen imports in 2030. In these strict visibility scenarios, hydrogen import remains the dominant supply route, reaching 300 TWh per year by 2045 (see Supplementary Fig. 8). In contrast, in the base scenario, domestic production of green hydrogen remains the primary source of hydrogen supply across the modeled years, with the exception of 2045, where the cost-optimal hydrogen import of 260 TWh per year would be required.

## Discussion
We developed a framework of analysis that enables a priori integration of visual impact considerations into techno-economic energy system planning at a large spatial scale. The findings suggest that preventing opposition towards large-scale renewable energy by placing those technologies out of sight from areas with average scenicness and population density are very costly and might reduce the resilience of the German energy system (see Fig. 5). However, only excluding large-scale renewable energy that are visible from the most scenic or densely populated areas would not lead to a notable change in the energy system design and costs. This indicates that moderately considering visual impacts for renewable energy planning in Germany is not financially binding. In these moderate scenarios, there is a slight preference for deploying open-field PV with lower visual impacts over onshore wind turbines.

In the scenarios with strict visibility restrictions, in which onshore wind turbines and open-field PV are placed out of sight of areas with average scenicness level or population density, the energy system cost increases by up to 38%. In these scenarios, a significant adoption of offshore wind and rooftop PV is necessary to substitute open-field PV and onshore wind power. For rooftop PV, up to 18 times the current rooftop capacity (618 GW) would be needed by 2045. This would necessitate an expansion rate of 29 GW per year until 2045, exhausting the available rooftop potential[56]. It is questionable whether the current expansion rate for rooftop and open-field PV in Germany of only 7.5 GW per year could be increased so much. Furthermore, to assume that all building owners would adopt rooftop PV is highly optimistic, as the current rooftop PV adoption still faces various socio-economic

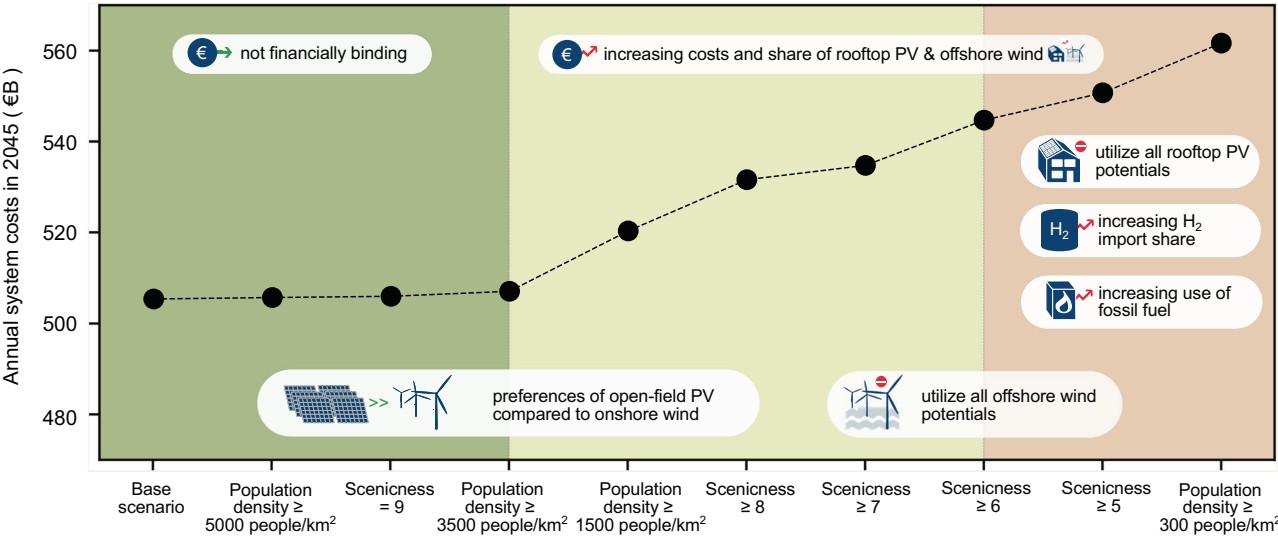

**Fig. 5 | Annual system costs in 2045 for different visibility scenarios.** The green color indicates scenarios where minimizing the visibility of renewable infrastructure exclusively from the most scenic or densely populated areas does not significantly affect system costs. By contrast, the beige color signifies scenarios where increasing visibility restrictions lead to a rise in system costs. The red color indicates the most stringent visibility restrictions considered, which results in cost escalation, exhaustion of renewable energy potential, and increases in hydrogen imports and fossil fuel reliance. As all sector's costs are taken into account, these system costs include many parts, such as for the building stock or transport options, which are not as strongly affected by the various scenarios as the energy sector. Source data are provided as a Source Data file.

challenges due to high upfront investment[61,62], unclear ownership and benefit-sharing schemes among landlord and tenant[63,64], and low dissemination of information about the technology[61,62]. For instance, Switzerland has applied a strategy to restrict open-field PV due to concerns over landscape visual impacts and prioritize rooftop PV[65]. This approach, however, has been criticized for being neither cost-effective nor spatially efficient[65] in the current policy settings. To overcome this, it is necessary to first address market barriers through the provision of subsidies targeted at low-income households. This will make rooftop PV more economically viable and encourage the adoption of this technology among first-time adopters[66]. Furthermore, advancing policy to support the collective self-consumption framework in multi-family buildings[64] and to regulate solar obligation on new public buildings may also help in accelerating rooftop PV deployment at high visual sensitivity scenarios.

Placing renewable infrastructure out of sight, even from areas with average scenic quality and average population density level, would necessitate a significant share of green hydrogen imports from as early as 2025. This could encourage neighboring countries to use existing renewables in the countries to produce green hydrogen for export to Germany. This scenario may involve replacing domestic renewable energy supply with other non-renewable sources, potentially leading to increased emissions in exporting countries[67]. Other strategies, such as promoting flexibility in hydrogen supply chains through the development of domestic production hubs, may reduce dependence on hydrogen imports. However, increased domestic production of green hydrogen would add significantly to the overall system costs. At this high visual sensitivity scenario, the few available renewable energy plants available for hydrogen production are located in high-levelized cost locations, as shown in Fig. 3. Other more cost-effective strategies that could be explored to reduce the reliance on green hydrogen imports include tapping into domestic biomass potential and implementing demand-side management through building retrofits to reduce electricity demand.

High visual sensitivity to renewable infrastructures would also require renewable infrastructures to be sited mainly offshore, or in certain remote areas, as shown in Fig. 2. This may concentrate environmental burdens in the locations not visible from scenic or populated areas and further give rise to concerns about distributive justice, given the potential for spatially unequal benefits and burdens associated with such strategies[68,69]. However, it is important to note that the spatial distribution of local benefits and burdens depends not only on where renewable infrastructure is located, but also on how the benefits of renewable energy installations are redistributed[70]. This includes considerations such as local ownership structures and access to affordable renewable energy. To facilitate a just transition, it is essential to understand what is perceived as just by the community. The reverse viewshed method demonstrated in this study can be applied at the local level to incorporate local preferences regarding the visibility of renewable infrastructure in the planning process. While the hidden placement of renewable energies demonstrated in this study may be a strategy from the spatial planners' perspective, this approach should not preclude public participation in the decision-making process. Rather, it should be used as a means to reflect public preferences in the planning process.

Our findings align with those of previous studies that suggest the implementation of moderate exclusion zones to account for the disamenities caused by renewable energy infrastructures[71,72]. We added to the analysis by providing exclusion zones based on visibility from scenic and populated areas at different sensitivity levels. Our findings indicate that moderate consideration of visual impacts in German renewable energy planning does not affect energy potential, optimized costs, and energy system transformation design. However, more stringent restrictions would result in significant output effects[72] through the utilization of more sites by other, more costly technologies. Furthermore, given the variability in population density and scenic areas across the country, a one-size-fits-all policy approach may be ineffective in implementing concrete policy at the local level. The results we present here show that there are some no-regret locations for renewable energy installations that are neither visible from scenic nor densely populated areas. Local governments could develop guidelines that vary according to local scenic and population

characteristics to ensure that renewable energy development aligns with both energy and aesthetic priorities. For example, by prioritizing buffer zones based on viewsheds from scenic or population thresholds (e.g., sceniceness = 9 and population density ≥ 3500 people per km$^2$), policymakers can achieve a balance between preserving aesthetic value and promoting renewable deployment. In addition, policymakers could incentivize offshore wind or rooftop PV projects in visually sensitive areas through measures such as expedited permitting[73] or tax breaks[74].

The impact of placing renewable energy infrastructures in locations that are not visible from important landscapes on energy system costs and design may vary by country or region, contingent on the spatial correlation between renewable resources and the viewsheds from important landscapes. For example, in Great Britain, areas with high-quality wind resources coincide with scenic areas[25]. In this case, excluding onshore wind visible from scenic areas might be more costly than in Germany. In addition, the selected visibility threshold distance, the availability of land area in the country, and the affordability of substitute energy resources may affect the magnitude of impact on system costs and the feasibility of minimizing the visual impacts of renewable infrastructure under greenhouse gas (GHG) neutral targets. In the reverse-viewshed analysis conducted in this study, the lowest visibility threshold distance is utilized, assuming that only major landscape changes from renewable energy are undesirable. Assuming a higher visibility threshold distance would further exclude land for renewable energy and may significantly increase the overall cost. It would also be interesting to investigate the impact of minimizing the visual impact of renewable infrastructures in other countries with limited land areas and constrained energy resources.

While the scenarios presented here are based on empirical studies that indicate that the dominant visibility of renewable energy on scenic landscapes is not preferred and leads to rejection[24-29], it is important to note that the visibility of renewable energy infrastructure is not always universally perceived as a negative visual impact by the public. For instance, in the case of wind energy infrastructure, perceptions of wind turbines can range from fully positive ones originating from feelings of progress and sustainability to fully negative ones induced by a critique of landscape industrialization[30,75]. Moreover, even negative perceptions are not always a direct indication of actual willingness to oppose projects, since it has been demonstrated that opposition to projects is often led by dedicated vocal minorities[13,26,76], other than by institutional means such as from administrative or voted local authorities[30]. In addition, the findings presented here illustrate that reducing the visibility of renewable infrastructures to the greatest extent alone would be an expensive strategy for increasing acceptance. This underscores the significance of combining the visibility consideration in planning with other efforts to enhance local acceptance of renewable energy projects, such as by ensuring local involvement in the planning process[15,77] and providing ownership schemes for local communities[78].

The framework utilized in this study is applicable to other regions, provided that the selection of landscape importance is adjusted to align with the specific needs of the region of interest. This could serve as a tool to facilitate a two-way dialog between national planners, local stakeholders, and the public with the goal of jointly planning the acceptable deployment of renewables[79,80] while understanding the implications of different planning arrangements on the overall energy transformation and system costs.

## Methods
### General approach
Our methodology is divided into four steps (see Fig. 6). First, reverse viewshed maps were calculated from 357,588 viewpoints representing every kilometer-square of Germany, on the EU-DEM v1.1 elevation model (with a grid resolution of 25 m[55]). These viewsheds maps have

underlying metadata, in our case, population numbers and landscape sceniceness ratings from 1 (low sceniceness) to 9 (high). Second, the influence of viewsheds on renewable energy potentials of onshore and offshore wind power and open-field solar PV were evaluated. This was done by excluding renewable energy plants visible from different populations and sceniceness thresholds utilizing the generated reverse-viewshed maps. Using ETHOS.GLAES[56,81,82], the generated reverse-viewshed maps were combined with other regulatory, geographical, technical, and environmental land eligibility constraints for siting utility-scale wind turbines and solar PV in Germany (see Supplementary Table 1). Third, we used ERA5 data to simulate electricity generation for different visibility scenarios using the ETHOS.RESKit tool[82]. Finally, the remaining renewable energy potentials that are not visibile from different sceniceness and population density thresholds were used to determine the impact of the visibility restrictions on the cost-optimal, greenhouse gas-neutral energy system transformation in Germany by 2045. The energy system optimization was conducted using the model ETHOS.NESTOR, which is based on the ETHOS.FINE modeling framework[83,84].

### Reverse-viewshed analysis
Reverse-viewshed analysis is a viewshed analysis conducted from the perspective of the landscape elements that are to be protected, rather than from the perspective of the proposed renewable energy projects. This analysis enables a priori visual impact assessment of renewable energy infrastructure[51]. It uses user-defined important landscapes (i.e., scenic areas, densely populated areas, or protected landscapes) as viewpoints and generates a reverse-zone of theoretical visibility (R-ZTV) map or reverse viewshed map that illustrates the areas in which any future construction of renewable energy infrastructure, would be visible from important landscapes. The generated R-ZTV map can be used as an exclusion zone in renewable energy planning to minimize the visual impact of renewable energy infrastructure on important landscapes.

We performed the calculation of R-ZTV using the r.viewshed function in GRASS-GIS[85]. The inputs required for this analysis are a high-resolution digital elevation model (DEM), viewpoints representing important landscapes, renewable energy infrastructure heights, observer height, and visibility threshold distance. For DEM data, the digital surface model (DSM) raster file for Germany from Copernicus EU-DEM v1.1[55] with 25-m resolution was used. The viewpoints used are every centroid of each km$^2$ of Germany, with the underlying metadata of sceniceness ratings[57] and the population density data[86]. A total of 357,588 viewpoints representing the centroids of each km$^2$ of Germany was utilized.

For the parameters used in this analysis, we set the observer height to 1.8 m, to reflect the upper bound of eye-level height of people in Germany, according to human body dimensions data for ergonomics standard[87]. This is to account for areas with bare-earth elevation height. We set a utility-scale wind turbine hub height (target height) of 130 m, based on an expert survey[88] of expected wind turbine hub height in 2035. The hub height is employed as the target height instead of the total height to prevent overestimation of visibility, due to different angle of views of the observer[59] and visibility at night time that depends on lights placed on top of the turbine hub. For the solar PV height, we used a height of 2 m.

The maximum distance of visibility, also referred to as the visibility threshold distance[89], exerts the most significant influence on the results of the R-ZTV analysis, as it defines the radius of application of the viewshed analysis[51]. In visibility analyses for large spatial scales, the utilized threshold distance ranges from 10 to 35 km[30,90]. When limiting the visibility effect to dominant visibility within a landscape, the range of threshold distance are reduced to 2 to 8.1 km[91,92]. In our analysis, we utilized a visual threshold of 10 km, and extended it by an additional 1 km, reaching 11 km. This adjustment accounts for the additional distance from the grid border to the viewpoint in the center of each km$^2$ of the sceniceness grid. This lowest edge of the spectrum for

visibility threshold distance is selected for two reasons: First, as the R-ZTV maps generated in our analysis are proposed as exclusion zones, a stricter definition of visibility is deemed appropriate. Secondly, it is acknowledged that visual impact is inherently a matter of subjective landscape perception, thus it embodies important uncertainties. These uncertainties refer to both current differences over perceptions[26,30,75,76,90,93] and also to the potential for positive-negative shifts in perceptions over time[13,77,94,95]. Consequently, although calculations of visual impacts are valuable, their integration in planning should be carefully considered and responsive to evolving social preferences over time. R-ZTV is an effective approach in this regard, as the generated maps can be utilized in the future both as exclusion zones or as weighted spatial layers in multi-criteria studies that consider local preferences at a smaller planning scale. For solar PV, the selection of a visibility threshold was more straightforward, as there are only a few studies that referred to visual impacts from solar energy. Therefore, the radius used in the study by Palmer[96] was adopted: 6.4 km, extended by 1 km, and rounded to 7.5 km, to account for the additional distance to the viewpoint in the center of each $km^2$ of the scenicness grid. Furthermore, additional considerations such as earth curvature and atmospheric refraction were omitted from the analysis as it requires more computational power. The exclusion of earth curvature consideration from the analysis was an additional rationale for selecting a lower-end for solar PV height. This was done to counterbalance the potential for overestimation. Consequently, a height of 2 m was selected for open-field PV panels, which typically range from 2 to 5 m in agrivoltaics applications[97,98].

Due to the large number of viewpoints analyzed in this study, which requires considerable computational power, the calculation of reverse viewsheds was performed on a high-performance computer cluster. The analysis was performed in parallel on several computing nodes for each county in Germany using GRASS-GIS version 7.8[85]. This software was chosen because it is open-source, relatively lightweight, and its functions can be accessed without explicitly starting the program, making it easy to use in scripts and batch processes.

After generating individual reverse-viewshed map from each centroid of 1-$km^2$ grid cell in Germany, the reverse-viewshed maps of viewpoints that meet the scenicness level or population density thresholds of each scenario (scenicness = 9, ≥ 8, ≥ 7, ≥ 6, ≥ 5, or population density ≥ 5000 people per $km^2$, ≥ 3500 people per $km^2$, ≥ 1500 people per $km^2$, ≥ 300 people per $km^2$) were merged using the r.series function in GRASS-GIS. The merge process combines all reverse viewshed raster files from viewpoints that meet the thresholds with a union operator at a resolution of 25 m. The merged R-ZTV maps from each scenario were then used as exclusion zones in the land eligibility and potential analyses.

## Land eligibility and renewable energy potential assessment

The merged R-ZTV maps for different visibility scenarios were incorporated into a land eligibility assessment as an additional exclusion zone. The open-source framework ETHOS.GLAES[81,82], which is based on the Geospatial Data Abstraction Library (GDAL)[99] was employed for this assessment. ETHOS.GLAES allows binary land exclusions in the study region based on user-defined criteria. The base scenario employs multiple high-resolution (as fine as 10 m) land exclusion maps, including legal, geographical, technical, environmental, and cultural preservation restrictions (see Supplementary Table 1) in accordance with the potential analyses previously conducted by ref. 56. The reverse viewshed maps are then combined with other land exclusion datasets in the land eligibility assessment to calculate the remaining renewable energy potentials at different visibility scenarios.

The base scenario for onshore wind includes forest and protected landscapes as eligible areas[56]. The ETHOS.GLAES algorithm subsequently places onshore wind turbines on the remaining eligible area while maintaining a spacing distance between turbines of eight times

the rotor diameter parallel to the main wind direction and four times the rotor diameter perpendicular to it[56,82]. A reference onshore wind turbine size of 130-m hub height, 174-m rotor diameter, and 5.5-Megawatt (MW) capacity was assumed[88]. However, as the average wind speed at each federal state varies across Germany, different wind turbine sizes were utilized for each federal state, reflecting the optimal size[82]. Electricity generation simulations were then conducted using ETHOS.RESKit[82], employing the ERA5 reanalysis data, with an assumed 15% loss factor to account for losses due to wake effects[100].

The base scenario for offshore wind turbines excludes military areas and has a 15-km buffer distance from the coastline. Future offshore wind turbine designs with a capacity of 17 MW, a hub height of 151 m, and a rotor diameter of 250 m were utilized[88]. The same rotor diameter distances were maintained between turbines as for onshore wind, and a 15% loss factor was assumed.

Finally, to derive the eligible area for placing open-field PV in the base scenario, we combined areas with low soil quality and 500-m-long shoulder strips of motorways and railways, reflecting the subsidy areas in accordance with the latest Renewable Energy Sources Act[60]. A capacity density of 79.2 MW per $km^2$ was assumed[56]. ETHOS.RESKit was also utilized to simulate the electricity supply of solar PV at the eligible sites for each visibility scenario utilizing the ERA5 reanalysis data.

## Energy system optimization

The ETHOS.NESTOR energy system model[101–103] serves as one of the foundations for the present analyses. This optimization model is based on the ETHOS.Fine modeling framework[83,84] and maps the national energy supply from primary energy to final energy across all potential paths and technologies. The model enables the creation of normative scenarios for Germany's future energy system and provides information on cost-effective ways to reduce greenhouse gas emissions. The model uses a linear optimization approach and is implemented bottom-up, i.e., the model maps individual technologies and components of the energy system in detail. Geographically, the model is limited to Germany but considers imports and exports of energy sources. For the present analysis, we chose a 5-year investment period, and the time horizon is set up to the year 2045.

The objective function is to minimize the total annual discounted system costs, including both fixed and variable costs. The fixed costs comprise capital expenditure and fixed operating expenses. The capital expenditure for a specific technology is determined by disaggregating the total investment into constant annuities, employing the capital recovery factor over the assumed depreciation period of that technology (which may differ from its technical lifetime). Additionally, the model considers the costs associated with supply infrastructure, such as electricity, heating, and gas grids, as well as transport infrastructure.

In consideration of externally specified boundary conditions (e.g., greenhouse gas reduction targets) and assumptions (e.g., industrial goods production, transport demand), the most cost-effective combination of technologies and energy sources that simultaneously satisfies all constraints is determined. The transformation pathways are determined based on the cost-optimized hourly-resolved operating plans for all installed technologies. The cost-optimal pathways can be interpreted as the decision of a "central" planner and represent a macroeconomic perspective. The macroeconomic perspective ensures that the possible technologies or measures are not influenced by any external factors, as neither current tax nor subsidy mechanisms are considered. Consequently, the scenarios presented here represent cost-optimized energy transformation pathways that are not intended to forecast the future. Rather, they are designed to demonstrate what is theoretically possible. This approach ensures that the real cost-optimal transformation pathway is not biased by the potentially misplaced subsidies currently present in the German energy system.

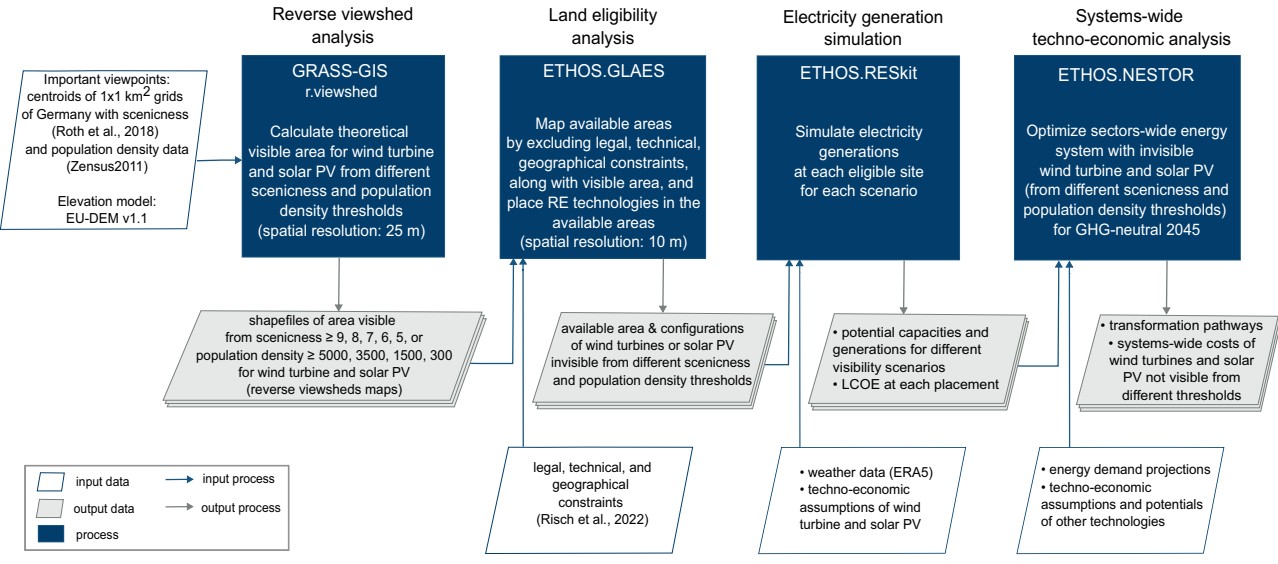

**Fig. 6** | Overview of the models and data used in this study.

The technologies that can be installed to meet the hourly energy demand and energy-related materials of all sectors are divided into generation, conversion, and storage technologies. The model includes relevant future technologies and measures (e.g., PV, wind power, heat pumps, short- and long-term storage, retrofitting of buildings, etc.). The feed-in from energy sources is limited by their time-dependent potential and efficiency (e.g., weather-dependent feed-in profiles for PV and wind power technologies). The installable capacities of the technologies have upper potential limits that are defined by technical restrictions[56] and, in some cases (PV and wind power) also by the visibility restrictions in the scenarios considered in this study. Energy conversion in power plants and other facilities is determined by their efficiency and capacity. The operation of energy storage systems is subject to restrictions in terms of charging and discharging rates and storage capacity.

A distinctive feature of the model is that all potential GHG-reduction options across all sectors (i.e., energy, transport, buildings, industry) are integral in competition with each other. One of the most important boundary conditions of this study are the exogenously overarching greenhouse gas reduction targets set by the current Climate Change Act. This target path must be adhered to by the model in any case (i.e., at every hour of every day from 2020 to 2045). The technologies and sectoral emission reduction contributions that can be utilized to achieve this target are the result of cost optimization. This means that important energy consumption, such as electricity consumption or hydrogen demand, are not exogenously assumed and predetermined, as is often the case in many other studies. Instead, it results from the diverse cost-optimized combination of different technologies and their use.

In the case of hydrogen, for example, this means that no specific hydrogen demand is assumed for individual years but that specific energy services are extrapolated into the future (e.g., the amount of steel produced in Germany or the tonne-kilometers traveled by trucks in Germany). These services can be provided via different routes within the model (e.g., crude steel production via the coke-fired blast furnace or with hydrogen in direct reduction plants, or the use of diesel trucks compared to fuel cell trucks). The model selects the option that can fulfill the respective service in the most cost-effective manner and meet all externally set constraints (e.g., greenhouse gas reduction targets). The model does not consider cost-effectiveness on a sectoral basis; rather, it assesses the cost-effectiveness of the entire system. This not only guarantees a cost-optimal solution at the system level, but also makes it possible to analyze the system benefits of the various solutions. In the case of hydrogen, the possibility of long-term seasonal storage should be

mentioned here in particular, which makes it possible to operate the energy system robustly even during cold dark doldrums. The requisite scale of hydrogen storage facilities and the proportion of renewable electricity to be converted into hydrogen for optimal operation of renewable energy plants can be assessed by the integrated energy system model ETHOS.NESTOR, which optimizes the use of the hydrogen model endogenously. The model has been described, used, and extended in numerous studies[101–105].

## Limitations and outlook

While this study has expanded upon earlier works[44,51] by incorporating visual impact considerations into techno-economic analysis and at a substantially larger planning scale, a number of limitations should be acknowledged. Firstly, a nationwide scenicness dataset utilized to generate reverse viewshed maps is, to our knowledge, currently only available for Germany and Great Britain. To enable similar assessments in other regions, proxies of landscape importance or scenicness would be required. This could be locations of national parks, heritage sites, or other designated important landscapes and landscape components[54]. However, even such datasets may not be available for every country and for all types of landscape components[51]. Alternatively, future studies could use public-selected landscapes of importance or characterize areas where opposition due to visual landscape impacts of renewable energy infrastructure has been documented in the past.

Secondly, the scenicness dataset employed as viewpoints in the reverse viewshed analysis has a relatively low resolution of 1 km. This may result in the omission of heterogeneity in landscape quality within a 1-km² area. Nevertheless, it should be noted that the centroids of the 1-km² grid are employed solely for the purpose of sampling viewpoints across Germany with diverse landscape quality. The impact on the resulting reverse viewshed map is anticipated to be minimal. This is because of the fact that the resulting viewshed map is largely contingent upon the spatial resolution of digital elevation map employed and the visibility threshold distance used. The present study used the EU-DEM v1.1, which is capable of capturing small-scale features at 25-m resolution. Additionally, as stated above, the assumed visibility threshold distances have been adjusted to account for the additional distance to the viewpoint in the center of the grid. Future studies that aim to conduct a more detailed analysis at a smaller spatial scale could benefit from interpolating the scenicness dataset to increase the number of viewpoints.

Thirdly, the input of the reverse viewshed analysis employed in this study is a digital surface model (DSM), which captures both natural

and artificial features, e.g., buildings and trees. At locations with a high number of features, such as forests and cities, the reverse viewshed obtained might be overestimated since the observer point is assumed to be on top of these features. When viewed from the ground, the resulting viewshed area in urban settings could be lower than the results obtained from the present analysis. However, due to the large geographic scale of this study, the deviation is negligible. A sensitivity analysis (see Supplementary Fig. 14) indicates that the calculated eligible area exhibits minimal variation (0–0.5%) when either DSM or digital terrain model (DTM)[106] is employed in the reverse viewshed calculation. A slight difference in grid size between the only available DTM raster (20 m) and DSM raster (25 m) may also contribute to this variation. A detailed viewshed analysis with much higher spatial resolution could combine the DSM with a DTM, which is a bare-earth elevation model, to detect possible features blocking the line of sight. This improvement could be beneficial for a micro-level (i.e., municipal) analysis where the exact locations of power plants need to be determined. This approach is, however, constrained by the current availability of DTM data, which is only accessible for selected regions. Additionally, a uniform 130-m turbine hub height was assumed in the reverse viewshed calculation, despite the use of varying optimal wind turbine heights (i.e., hub height of 130–170 m) for each federal state in the potential analysis. Nevertheless, the variation in the resulting eligible area remains minimal at 0–0.2% (see Supplementary Fig. 15). Additional limitations and uncertainties originate from the setup of the viewshed analysis per se, which does not address parameters such as the exposure of visible items[92], contrast[89], and angle of viewing[107]. The reverse viewshed analysis employed in this study did not consider the effects of earth curvature and atmospheric refraction. This resulted in an overestimation of the visible area for 2-m solar PV by 1–2%. This effect is lower for larger objects such as onshore wind turbines. Future studies could consider earth curvature and atmospheric refraction in the viewshed analysis to improve accuracy.

In our reverse viewshed analysis, we quantified the potential visual impact of a single wind turbine on multiple important landscapes and used it as an exclusion zone in the land eligibility model. This would result in the placement of any number of wind turbines out of sight of multiple important landscapes considered. A limitation of the current settings is that it is not feasible to assess the land suitability based on the potential visual impact of single or cumulative numbers of wind turbines. This is because the necessary information regarding turbine locations and numbers must be known in advance of the energy system optimization and would be more appropriate for a smaller-scale analysis. Future studies for the detailed design of a wind energy project, could employ cumulative reverse viewshed analysis and count the number of planned wind turbines within the combined reverse-zone of theoretical visibility (R-ZTV). Each turbine could then be weighted by the number of viewpoints it affects.

In our analysis, we integrated the grid expansion into the infrastructure costs within the optimization model. However, we did not consider the visual impacts of grid construction. The substantial deployment of large-scale renewable energy would require grid development, which may also give rise to local opposition due to their visual impacts on the landscape[108] and indirect environmental effects[38]. Future studies might employ a spatially explicit optimization model to simultaneously exclude cable routing with high visual impacts. Additionally, the potential impact of climate change on renewable energy generation was not modeled in the present study. Climate change-induced extreme events in the future may significantly affect renewable generation, particularly for wind power in some regions[109]. To model this accurately would require transdisciplinary collaboration between energy systems and climate modelers, which currently still tends to be limited[110]. Future studies could incorporate climate-related uncertainty analysis in addition to the past meteorological data used for the simulation.

Furthermore, while the investigation of visibility from scenic and densely populated areas partially quantifies the visual impacts of infrastructure on important landscapes[42], not all aspects of visual impacts[51] are included in the present analysis. The concept of visual impacts also depends on the level of detection, recognition[111], perceptions of annoyance from renewable energy infrastructure, and is also influenced by place attachment[112]. Over the past decade, it has become evident that the perception of visual impact from renewable energy infrastructure is also not a strictly independent criterion for individuals, as it is intertwined with other parameters that affect local acceptance. Such factors may include community participation in the design and planning process[77], the application of landscape studies in the planning process[94], the pre-existing character of the landscape[95,96], or even the professional backgrounds of the affected populations[13]. Acceptance of large-scale renewable infrastructure may also change over time[113] and be influenced by local people's familiarity and experience with renewable energy projects[114]. This issue would be complex to model, as in our current study, the renewable energy plants at the various locations are only input data prior to optimization, and it is not clear which sites would be selected in which investment period based on cost optimality or other criteria. To reflect this, it would be conceivable in future studies to develop an iterative approach that assesses the optimization results considering the evolving societal preferences depending on the number of installed renewable plants in a given region and then adjusts the input data. Additionally, site-specific visual impacts, such as shadow flicker phenomena from wind turbines[115], glare from PV[116], or varying severity of visual impacts due to a cumulative number of concentrated renewable energy plants[75], could be incorporated in future analyses.

## Data availability
Source data for Figs. 2–5 are provided in the Source Data file. The land use data that support the analysis of this study are partially publicly available as listed in references of Supplementary Table 1. However, some data are restricted due to data protection. For example, the high-resolution building data from the German Federal Agency for Cartography and Geodesy (BKG), which were used under license for the current study. Source data are provided with this paper.

## Code availability
All models used for this study are open source. The code developed for the large-scale reverse viewshed analysis can be accessed on Jülich DATA[117]. The software tool ETHOS.GLAES[118] is used for land eligibility analysis. The ETHOS.RESKit tool[119] is employed for the electricity generation simulation. Finally, the ETHOS.FINE energy system optimization framework[120] is used to instantiate a national energy system model utilized in this study. We are currently also working on an open-source publication of this national energy system model. In the meantime, please contact the authors for more information on the model setup if not included in this manuscript.

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

## Acknowledgements
We would like to thank Maximilian Hoffmann for his constructive feedback on the preliminary version of this manuscript. The Helmholtz Association's program "Energy System Design" supported T.T., T.P., R.M., S.R., F.K., D.S. and J.M.W. for this work. Additionally, funding from the European Union's Horizon Europe research and innovation program under grant agreement no. 101083460 (WIMBY) supported R.McK. and R.C. for this work. We also express our gratitude to the Bodossaki Foundation, which provided support to R.I. for this work.

## Author contributions
Conceptualization, J.M.W., R.I. and T.T.; methodology, T.T., T.P., J.M.W., R.I., R.M., S.R. and R.C.; formal analysis, T.T., T.P., R.M., and S.R.; data curation, J.M.W., T.T. and T.P., writing—original draft, T.T., J.M.W., R.I., F.K. and R.C.; writing—review and editing, J.M.W., F.K. and R.McK.; writing—interactive feedback, R.McK. and D.S.; visualization, T.T., T.P., R.M., R.McK. and J.M.W.; funding, acquisition D.S.

## Funding

## Competing interests
The authors declare no competing interests.
