## [Transparent Peer Review file · Nature Communications]

Quantifying the trade-offs between renewable energy visibility and system costs

Corresponding Author: Ms Tsamara Tsani

Version 0:

Reviewer comments:

Reviewer #1

(Remarks to the Author)

The paper addresses the visual impact of nationwide renewable energy systems, focusing on the case of Germany. It combines viewshed and techno-economic analyses to assess how siting renewable energy sources (such as wind turbines and photovoltaic plants) out of sight from scenic and densely populated areas could affect the cost, design, and resilience of the energy system. The results show that moderate restrictions based on visual impacts do not significantly raise system costs, but higher sensitivity to visual impacts can increase annual system costs by up to 38%, with a heavier reliance on hydrogen imports and rooftop photovoltaics. The paper suggests that while the visual impact of renewable infrastructure cannot be entirely avoided, strategic planning could allow for socially acceptable renewable deployment without heavily compromising system cost-effectiveness. Major revision is recommended due to the following reasons and comments:

1. The paper's analysis is confined to Germany. How would the methodology and findings translate to other regions or countries with different geographic and socio-economic conditions? A broader comparative discussion could strengthen the generalisability of the results.
2. In the introduction, it was mentioned that the use of clean fuel from renewables is a practical solution, which is fine. Have the author initially considered the impact of climate change scenario on the progress of the energy sources and, particularly renewables, as in the recent work of "Cost, environmental impact, and resilience of renewable energy under a changing climate: a review. *Environ Chem Lett* 21, 741–764 (2023). <https://doi.org/10.1007/s10311-022-01532-8>.", the important question is what the impact of climate change will be on each and every renewable source, with solar, wind, geothermal, hydropower, and biomass being the most common. That recent publication provides insight into the impacts of climate change on renewable energy sources and their future prospects under climate change scenarios, which will add real value to the revised manuscript to mention this and discuss it within the revised manuscript.
3. The reliance on hydrogen imports and rooftop PVs in high-visual-sensitivity scenarios is acknowledged, but what are the potential technical and social barriers to mass adoption of rooftop PVs in Germany, especially in urban areas? Addressing these barriers would make the conclusion on resilience more credible.
4. The paper assumes that the visual sensitivity of the population remains constant until 2045. Have the authors considered how evolving societal views on renewable infrastructure might affect public acceptance over time? A sensitivity analysis that accounts for changes in societal preferences would be valuable.
5. The authors mention that the adoption rate for rooftop PVs is much lower than required for the strict visibility scenarios. Could the paper expand on possible policy measures or incentives that might close this adoption gap? This would add practical relevance to the proposed solutions.
6. The exclusion zones based on visual impacts largely focus on scenicness and population density. However, are there other social or environmental factors, such as biodiversity or historical preservation areas, that could further constrain renewable energy deployment? Including these dimensions would make the planning framework more comprehensive.
7. There is minimal discussion on the economic or environmental impacts of the grid expansions required to compensate for the reduction in large-scale renewable infrastructure. Could the authors incorporate this into their cost analysis, as grid expansion could also face significant opposition due to visual or environmental concerns?
8. The renewable energy integration and social impact and particularly in electricity generation is not well written and the authors could refer to this work "Social, environmental, and economic consequences of integrating renewable energies in the electricity sector: a review. *Environ Chem Lett* 21, 1381–1418 (2023)." to strengthen this part.
9. The reverse viewshed analysis is innovative, but more details on the limitations of this method (e.g., accuracy, computational cost) are needed. How do uncertainties in this analysis affect the overall system cost predictions?
10. In terms of policy implications, the paper suggests that moderate visual restrictions do not heavily affect system costs.

However, what specific policy recommendations can the authors offer for local governments balancing the need for renewable energy with visual and environmental concerns? Concrete examples would improve the paper's practical utility.

11. The reliance on hydrogen imports in high-sensitivity scenarios raises concerns about energy security. Could the authors explore alternative strategies to mitigate the reliance on imports, such as increased domestic production or enhanced storage technologies?

12. The results indicate that excluding renewables from scenic and densely populated areas would lead to a more concentrated deployment in less visible regions. Has the paper considered the potential environmental justice issues, where the burdens of renewable energy infrastructure might disproportionately fall on less populated, rural areas?

Reviewer #2

(Remarks to the Author)

The main interest of this paper is that it presents, in a reasonably quantified manner, the cost of making wind and large scale PV energy compatible with the scenic value of a landscape. This is a highly topical issue, with public opinion controversies in almost every country. Since a large fraction of the interest, and merit, is the quantified analysis, an obvious shortcoming is that the results are valid for only one country, viz Germany.

However, even if taken as an example, the solid quantification (the flaws reported below notwithstanding) make the paper a relevant, significant contribution to the debate on the visual impact of renewable energies.

The modelling work can be thought of as consisting of two parts: the viewshed model, which determines the visually-affected areas, and the energy-economic model, which determines the impact in the economy of different renewable-energy scenarios.

The viewshed model is sufficiently described and appears to be robust. The main weakness is that the model spatial resolution (1 km²) is arguably insufficient to capture many small-scale features that would block the view, and hence perhaps allow a greater installed capacity in each scenario (particularly under high-population-density restrictions). This is briefly (and not very clearly) acknowledged by the authors in the Limitations section. But the way that this issue is addressed and discussed minimises the problem. The authors appear to argue (line 464 and following ones) that this effect is negligible because the difference between using a DTM and DSM is very small (0-0.5% of the eligible area). I would argue that this small difference is perhaps the proof that the relatively large resolution (of 1 km) is not adequate to capture this effect of small-scale features (which are under 1 km) blocking the view. While I acknowledge that the use of smaller resolutions may not be feasible for a study at the national scale, I think that the authors may be downplaying this effect, and perhaps they should clearly warn about it early in the paper.

The energy system model is only loosely described in the paper. This is of course a very complex model, and a detailed description is perhaps impractical. The authors refer to three publications (refs 83, 84, 85) where the model "has been described, used, and extended". But, as with this paper, the description in those references is partial and insufficient, and very much tailored to the application in the respective paper. Given the relevance of the energy model in the present paper, an expanded description of the main hypotheses, features and methods of the model would be advisable. If space is a limitation, this could be submitted as supplementary material.

The ease of interpretation of the methodology and results is at times marred by convoluted, vague or poorly thought-out presentation (which goes beyond a correct English usage). The authors should carefully review the manuscript so that the text is easy to understand on a first reading. Examples of instances where better language is needed are (partial list):

- The caption of Fig 1, in particular subfigure b
- The interpretation of Fig 2 (in particular, the fact that the horizontal axes move through the several scenarios on top of the graph)
- Supplementary figures 1 and 2 are in my opinion irrelevant for the goal of the paper
- Figures like Supplementary Fig 3 would benefit from a legend, in each subfig, for ease of interpretation
- The meaning of placement density in Supplementary Fig 3 is not clear, and hence the conclusion ("The existing placement density is observed to decline with an increase in scenicness level") is hard to validate
- The meaning of "sectors" in Fig 4a, Supplementary Fig 5 and similar ones is unclear: it seems to heterogeneously mix economy sectors (eg Transport or Energy) with others that may be cost categories (eg, Conventional Fuels or Renewable Fuels, which are part of the energy sector)
- The caveat in line 442 and following is vague and poorly expressed.

The authors need to critically examine the readability of the paper and edit it with clarity in mind.

Under the section "Data availability", the authors provide links to public repositories with codes and tools. The authors refer to the information provided as "frameworks", "code", "software tools". Inspection of the links provided appears to indicate that in some cases the software and data provided would allow to reproduce the paper results (this is perhaps the case of the

reverse viewshed analysis), but this is not so clear in other cases. For instance, the tool for land eligibility analysis appears to apply only to Aachen; and in the case of ETHOS.FINE it would appear that the software link is for the general framework, rather than for its instantiation for this work (eg with the relevant data).

Lastly, while English usage should not be addressed in this review (and it is not), because of its pervasiveness in the paper, I need to stress that scenicness is not, I think, an English word, and it is an ugly invention. Perhaps the authors can consider alternatives, such as scenic value or scenic quality?

Reviewer #3

(Remarks to the Author)

A very interesting, timely and relevant manuscript, which addresses a problem that has not received sufficient attention: the cumulative, regional, potential impact by visibility and its influence on pathways of the energy system transformation. A few aspects of the methods applied, however, remain opaque in the present manuscript.

General discussion:

Visibility is not equal to visual impact, i.e. a location exposed to visibility is not necessarily perceived as an unsuitable location. I believe that Shang and Bishop (2000) have pioneered the distinction between detection, recognition and impact in their paper "Visual thresholds for detection, recognition and visual impact in landscape settings. However, while there is little empirical evidence on the degree of visual impact for cumulative turbines or population, the question is whether visual impact is really a problem for the residents of population centres or the visitors to scenic places, or is it impossible to relate the intensity of felt visual impact to the preferences of people, like Ladenburg (2015) discussed in "Does more wind energy influence the choice of location for wind power development? Assessing the cumulative effects of daily wind turbine encounters in Denmark"? Also, is local opposition not usually a matter of a small minority of residents? I believe the paper needs a more thorough discussion of this aspect.

A grid resolution of 1km may result in a significant error for line-of-sight analysis especially in hilly terrain, and also the site eligibility and LCOE calculations will have significant inaccuracy levels at this scale. The manuscript owes a discussion of the likely error induced by this coarse resolution. Data of at least 100m resolution are available, and computational effort should be manageable as well.

The results are presented in a meaningful way, and valuable conclusions are drawn. However, the main parameters such as the cut-off distance and the definition of what is scenic beauty and dense population are fixed without exploring any sensitivity that may be contextual to the problem of the acceptance of wind energy and the role that visibility plays in it. E.g. most literature agrees that ownership form, scale of installation, or procedural justice in planning wind farms has an equally high if not higher importance.

Specific comments

Introduction:

Chapter heading „Introduction“ is missing

L 38-39: Rather than "mitigating the intensification of climate-related extreme events" I would say "mitigating climate change" in general. Besides extreme events, climate change also drives a general degradation of the climate.

L 43-45: Social opposition due to lack of acceptance is perhaps the most important, but not the only reason for reduced growth of on-shore wind energy in Germany. Others are political issues on the state level, supply chain problems of the industry, and perhaps most importantly delays in the planning and permitting of new wind farms, but also road infrastructure etc. Please consider to revise the motivation accordingly.

L 213 ff: Please explain why rooftop PV requires more storage than in the base case. I believe the reason is a lower deployment of wind energy, but that should be made explicit.

Methods:

The reverse use of the viewshed method sounds intriguing. However, how do you deal with cumulative viewsheds, which in standard ZVI analysis is a measure of the quantity of likely visual impact, as it counts the number of viewpoints visible at each location? Does the visibility of a single wind turbine have the same weight as the cumulative impact by a large number of wind farms?

Besides errors in elevation grids, here partially considered by a comparison of DSM and DTM, the most significant source of error in viewshed analyses is the raster resolution of the DEM used (see e.g. Riggs and Dean, 2007). Please describe the method by which the native 30m (EU-LAEA) resolution of the EU-DEM v1.1 elevation model (which btw. is no longer supported via the link provided [76]) has been aggregated to 1km. For viewshed analysis, one could argue for particular requirements to resampling, as averaging may not be the most suitable. In any case, errors are induced, and the authors owe an account of the likelihood of errors induced by resampling the central data source.

The choice of a 1 km raster resolution for land eligibility seems to have been motivated by the high computational effort required, and perhaps also by the scenic value grid used. However, land use maps are available at a 10 m resolution, as Risch, Mayer et al. (2022) suggest; and since Germany is characterised, in many places, by a highly fragmented land use pattern, see Schug, F. et al. (2020), the choice of a resolution a factor 10,000 less would induce a significant error. Would it be possible to estimate this error? On the other hand, the land requirement of a single reference turbine is roughly 1 km², so

is the raster resolution chosen out of convenience for the analysis?

I would argue that the scenic value dataset by Roth et al. (2018) could be interpolated to higher resolutions without loss of validity, thereby increasing the chance to better accommodate the complexity of landscapes.

Census data (77): which data have been used? The native 100m grid must have been resampled; please describe how. Like for the other data layers, by resampling to 1 km² cell size the analysis sweeps over the complex texture of land use and settlement patterns in Germany. A discussion of the likely errors would be required.

Observer height and dealing with obstacles in land- and cityscapes: a uniform observer height is difficult to include, since especially in urban environments; people rarely see much of their surroundings (surrounded by buildings) while in rural settings, vegetation will obscure their view. Also, the scenic value map could assist here as there is (visually assessed) a high spatial correlation between high population density and low scenic beauty.

L 345: While standard open field PV has a height of about 3 m, agri-PV would at least be 4 m high, depending on the type. With an observer height of 2 m, the horizon distance is 5 km, hence Earth curvature becomes relevant.

L 358: it is not clear if boundary effects are included (by considering viewpoints up to 11 km outside the boundaries of counties to include the visual impact from turbines located in neighbour counties) and whether the inclusion of the boundaries would not lead to an unnecessary high computational effort because the total number of line-of-sight analyses increases significantly.

It is not clear why this process was computation intensive. I found it to be no problem to run the viewshed tools in QGIS and ArcGIS Spatial Analyst for a 100 m elevation grid (SRTM4 resampled to 100m) for about 3,700 viewpoints for an area of about 70,000 km² on a standard computer. Even including earth curvature and atmospheric refraction is a matter of minutes. Most likely this is owed to the heuristic applied in QGIS, but it seems to be effective to increase raster resolution in the present study.

L 366: please explain the merging of R-ZTV maps: is it a logical OR function or do you use another logic?

Version 1:

Reviewer comments:

Reviewer #1

(Remarks to the Author)

The comments were addressed fully; hence, I recommend the publication of the manuscript in its current version without any further changes.

Reviewer #2

(Remarks to the Author)

The authors have addressed all of my comments to the first submission. In most cases, these have been addressed by making (usually minor) modifications to the text to qualify or correct their original statement, rather than by addressing the underlying issue (eg open data availability). Nevertheless, with these modifications, I think that the amended text better reflects the extent, novelty and limitations of the work.

Reviewer #3

(Remarks to the Author)

The authors have provided comprehensive answers to the reviewers' questions and taken great efforts to improve the manuscript. All suggestions were taken very seriously and the provided revision of the paper has greatly improved its quality. I recommend the publication without any further revision.

Dear editor and reviewers of the manuscript “Out of sight, out of mind? Cost of minimizing visibility of nationwide renewable energy systems” with ID: NCOMMS-24-56230-T. We thank you for your time and your constructive comments to improve the manuscript. We confirm that we have addressed each comment in the revised manuscript as outlined below. Please find our answers to your comments in blue. Changes in the manuscript are shown in *italics*.

Reviewer #1 (Remarks to the Author):

The paper addresses the visual impact of nationwide renewable energy systems, focusing on the case of Germany. It combines viewshed and techno-economic analyses to assess how siting renewable energy sources (such as wind turbines and photovoltaic plants) out of sight from scenic and densely populated areas could affect the cost, design, and resilience of the energy system. The results show that moderate restrictions based on visual impacts do not significantly raise system costs, but higher sensitivity to visual impacts can increase annual system costs by up to 38%, with a heavier reliance on hydrogen imports and rooftop photovoltaics. The paper suggests that while the visual impact of renewable infrastructure cannot be entirely avoided, strategic planning could allow for socially acceptable renewable deployment without heavily compromising system cost-effectiveness. Major revision is recommended due to the following reasons and comments:

1. The paper’s analysis is confined to Germany. How would the methodology and findings translate to other regions or countries with different geographic and socio-economic conditions? A broader comparative discussion could strengthen the generalisability of the results.

Thank you for raising this important point of discussion. The methodology proposed in our study is applicable to other regions, provided that the viewpoints for reverse viewshed are adjusted to utilize important points or landscapes in the region of interest. In our case, we use scenicness datasets that are currently only available for Germany and Great Britain. Applications in other regions can utilize proxies of landscape importance, such as historical heritage sites, tourism sites, important landscape selected by the public, or even using locations with records of past opposition to renewables (if available). In general, the integration of reverse visibility maps into land eligibility analyses, and further into energy system models that we pioneered in this study can be applied to another region. We acknowledged this in the discussion and limitations subsection:

In line 375: *“The framework utilized in this study is applicable to other regions, provided that the selection of landscape importance is adjusted to align with the specific need of the region of interest.”*

In line 591: *“Firstly, a nationwide scenicness dataset utilized to generate reverse viewshed maps is, to our knowledge, currently only available for Germany and Great Britain. To enable similar assessments in other regions, proxies of landscape importance or scenicness would be required. This could be locations of national parks, heritage sites, or other designated important landscapes and landscape components [54]. However, even such datasets may not be available for every country and for all types of landscape components [51]. Alternatively, future studies could use public-selected landscapes of importance or characterize areas where opposition due to landscape impacts has been documented in the past.”*

Regarding the findings, we acknowledge that placing renewable energy out of sight of important landscape might have different impacts on the remaining renewable potential and energy system costs in other regions. This mainly relates to how spatially correlated the viewsheds from important landscapes are with the renewable energy resources. For example, in Great Britain, areas with high onshore wind resources coincide with scenic areas (McKenna et al., 2021). In this case, excluding onshore wind visible from scenic areas might be more costly compared to the case for Germany. The impacts can be checked for other regions in future studies using the framework of analysis that we have presented in this study. We added the following in the discussion:

In line 343: *“The impact of placing renewable energy infrastructures in locations that are not visible from important landscapes may vary by region, contingent on the spatial correlation between renewable resources and the viewsheds from important landscapes. For example, in Great Britain, areas with high quality wind resources coincide with scenic areas [25]. In this case, excluding onshore wind visible from scenic areas might be more costly than in Germany. In addition, the selected visibility threshold distance, the availability of land area in the country, and the affordability of substitute energy resources may affect the magnitude of impact on system costs and the feasibility of minimizing the visual impacts of renewable infrastructure under net-zero targets. In the present study, the lowest visibility distance is utilized, assuming that only major landscape changes from renewable energy are undesirable. Assuming a higher visibility distance further excludes land for renewable energy and would significantly increase the overall cost. It would also be interesting to investigate the impact of minimizing the visual impact of renewable infrastructures in a country with limited land area and constrained energy resources.”*

2. In the introduction, it was mentioned that the use of clean fuel from renewables is a practical solution, which is fine. Have the author initially considered the impact of climate change scenario on the progress of the energy sources and, particularly renewables, as in the recent work of “Cost, environmental impact, and resilience of renewable energy under a changing climate: a review. Environ Chem Lett 21, 741–764 (2023).

<https://doi.org/10.1007/s10311-022-01532-8>.”, the important question is what the impact of climate change will be on each and every renewable source, with solar, wind, geothermal, hydropower, and biomass being the most common. That recent publication provides insight into the impacts of climate change on renewable energy sources and their future prospects under climate change scenarios, which will add real value to the revised manuscript to mention this and discuss it within the revised manuscript.

Thank you for the suggestions. We have now discussed this in our limitations section:

In line 660: “Additionally, the potential impact of climate change on renewable energy generation was not modelled in the present study. Climate change-induced extreme events in the future may significantly affect renewable generation, particularly for wind power in some regions [109]. To model this accurately would require transdisciplinary collaboration between energy system and climate modelers, which currently still tends to be limited [110]. Future studies could incorporate climate-related uncertainty analysis in addition to the past meteorological data used for the simulation.”

3. The reliance on hydrogen imports and rooftop PVs in high-visual-sensitivity scenarios is acknowledged, but what are the potential technical and social barriers to mass adoption of rooftop PVs in Germany, especially in urban areas? Addressing these barriers would make the conclusion on resilience more credible.

Thank you. We have mentioned the economic and social barriers to mass adoption of rooftop PVs in the discussion section, in line 275: “Furthermore, to assume that all building owners would adopt rooftop PV is highly optimistic, as the current rooftop PV adoption still faces various challenges due to high upfront investment [61, 62], unclear ownership and benefit-sharing schemes among landlord and tenant [63, 64], and low dissemination of information on the installation the technology [61, 62]”.

Additionally, we now added the policy incentives to close the gap. First to make PV adoption more economic for low-income households by offering targeted subsidies. As they become more economic, the policy needs to address the barriers and market failures, e.g. through regulations to proscribe PV on all new buildings and promote clear policy for collective self-

consumption to solve the landlord-tenant dilemma hindering rooftop PV adoption in multi-family buildings.

In line 283: *“ To overcome this, it is necessary to first address market barriers through the provision of subsidies targeted at low-income households. This will make rooftop PV more economically viable and encourage the adoption of this technology among first time adopters [66]. Furthermore, advancing policy to support the collective self-consumption framework in multi-family buildings [64] and to regulate solar obligation on new public buildings may also help in accelerating rooftop PV deployment at high visual sensitivity scenario.”*

4. The paper assumes that the visual sensitivity of the population remains constant until 2045. Have the authors considered how evolving societal views on renewable infrastructure might affect public acceptance over time? A sensitivity analysis that accounts for changes in societal preferences would be valuable.

Thank you. Indeed, there are various studies demonstrating that people’s perception on renewable infrastructure might change over time (Wolsink, 2007) and might be influenced by familiarity and experiences with renewable energy projects (Windemer, 2023). This perception may also affect their visual sensitivity towards renewable infrastructure. We tried to address this by setting different scenarios that aimed to serve as sensitivity scenarios, ranging from the strictest (e.g., scenicness ≥ 5) to the most relaxed (e.g., scenicness = 9, when visual impacts are only unacceptable at the most scenic places). Unfortunately, due to the non-linear relationship between the number of installed renewable energy plants and societal preferences, we did not endogenously model the changes of visual sensitivity on the deployment decision of renewable energy at each investment time step. Future studies with spatially explicit energy system models could enable such endogenous integration. We now included this point in the limitation section and additionally added a discussion point on the dynamics of visual sensitivity of renewable energy infrastructure.

In line 679: *“The acceptance rate may also change over time [113] and be influenced by local people’s familiarity and experience with renewable energy projects [114], the dynamic changes in people’s sensitivity to visual impacts from renewable infrastructure may also be a source of uncertainty for landscape impact quantification. This issue would be complex to model, as in our current study, the renewable energy plants at the various locations are only input data prior to optimization and it is not clear which sites would be selected in which investment period based on cost optimality or other criteria. To reflect this, it would be conceivable in future studies to develop an iterative approach that assesses the*

optimization results considering the evolving societal preferences depending on the number of installed renewable plants in a given region and then adjusts the input data.”

5. The authors mention that the adoption rate for rooftop PVs is much lower than required for the strict visibility scenarios. Could the paper expand on possible policy measures or incentives that might close this adoption gap? This would add practical relevance to the proposed solutions.

Thank you. Please see our response to your comment 3 above.

6. The exclusion zones based on visual impacts largely focus on scenicness and population density. However, are there other social or environmental factors, such as biodiversity or historical preservation areas, that could further constrain renewable energy deployment? Including these dimensions would make the planning framework more comprehensive.

Thank you. Besides considering visibility from important landscapes, in our land eligibility analysis we have excluded nature conservation areas and important cultural heritage areas. The complete list of our exclusion zones can be found on Supplementary Table 1. We excluded protected natural areas, protected water zones, natural parks, and bird protected areas obtained from WDPA (The World Database on Protected Area) dataset; biosphere and farmland from BFN (German Federal Agency for Nature Conservation, Bundesamt für Naturschutz) dataset; historical sites from Open Street Map; camp sites, recreational areas, and cemeteries from Basis-DLM (official German land use dataset). This is based on a previous comprehensive land eligibility study conducted by our co-authors (Risch and Maier, 2022: doi.org/10.3390/en15155536). We have now added the following sentences in the beginning of the results section and in the method section to clarify the inclusion of these conservations and cultural exclusion in our land eligibility analysis:

In line 110: *“The base scenario for our land eligibility assessment follows the study of Risch et al. [56] that considers legal, technical, geographical, environmental, and cultural preservation restrictions as listed in Supplementary Table 1.”*

In line 483: *“The base scenario employs multiple high-resolution (as fine as 10-m) land exclusion maps including legal, geographical, technical, environmental, and cultural preservation restrictions (see Supplementary Table 1) from potential analyses conducted by Risch et al. [56].”*

7. There is minimal discussion on the economic or environmental impacts of the grid expansions required to compensate for the reduction in large-scale renewable

infrastructure. Could the authors incorporate this into their cost analysis, as grid expansion could also face significant opposition due to visual or environmental concerns?

Thank you. You are right, grid expansion could also face opposition due to visual or environmental concerns (Bertsch et al., 2016). We briefly discussed in the limitation section that we did not incorporate the visual impacts of grid expansion due to the current limitation of our complex one-node energy system model. However, we have considered the grid expansion cost in the infrastructure sector in our energy system model. At high-visual-sensitivity scenario, the energy systems become more decentralized with local rooftop PV. As we have stated in the result section: “In these strict scenarios, marginal reductions in the infrastructure sector costs can also be observed (Fig. 4a), as the distributed use of rooftop PV reduces the necessity for grid expansions.”

We now added some sentences in the limitation section discussing the potential visual impacts of grid expansions and make recommendations for future studies:

In line 654: “In our analysis, we integrated the grid expansion into the infrastructure costs within the optimization model. However, we did not consider the visual impacts of grid construction. The substantial deployment of large-scale renewable energy would require grid development, which may also give rise to local opposition due to their visual impacts on the landscape [108] and indirect environmental effects [38]. Future studies might employ a spatially explicit optimization model to simultaneously exclude cable routing with high visual impacts.”

8. The renewable energy integration and social impact and particularly in electricity generation is not well written and the authors could refer to this work “Social, environmental, and economic consequences of integrating renewable energies in the electricity sector: a review. Environ Chem Lett 21, 1381–1418 (2023).” to strengthen this part.

Thank you. We have now included the referred study in our discussion on limitations:

In line 656: “The substantial deployment of large-scale renewable energy would require grid development, which may also give rise to local opposition due to their visual impacts on the landscape [108] and indirect environmental effects [38].”

and in the introduction:

In line 59: “It is crucial to address these concerns, as the integration of renewable energy sources, may potentially result in adverse local impacts in social, environmental, or economic terms, if not planned with sufficient consideration [38].”

9. The reverse viewshed analysis is innovative, but more details on the limitations of this method (e.g., accuracy, computational cost) are needed. How do uncertainties in this analysis affect the overall system cost predictions?

Thank you. We appreciate the reviewer's feedback on this matter. As highlighted in the limitation section, uncertainties from reverse viewshed analysis can result from several factors. The most crucial one, for example, originates from the basis digital elevation model (DSM or DTM) that is used for the analysis. However, we have shown through our sensitivity analysis (Supplementary Fig. 14) that the variation in results remains low. We have now added other uncertainties that might result from the setup of the viewshed analysis per se, that does not account for exposure of visible items, contrast, and angle of viewing.

In line 635: *“Additional limitations and uncertainties originate from the setup of the viewshed analysis per se, which does not address parameters such as the exposure of visible items [89], contrast [89], and angle of viewing [107].”*

Another important factor would be the selected visibility distance parameter. In the present study we used the lowest visibility distance parameter for each technology (i.e., 11 km for wind turbines and 7.5 km for open-field PV), where within this visibility distance renewable infrastructure significantly alters the landscape. The reason for this is that the resulting viewshed map is used as an exclusion zone, so only infrastructures that significantly alter the landscape are excluded. A future study could use a higher visibility distance, but this would exclude renewables even more and significantly alter the design of the energy transformation, making the overall cost even higher. We have added this point to the discussion:

In line 348: *“In addition, the selected visibility threshold distance, the availability of land area in the country, and the affordability of substitute energy resources may affect the magnitude of impact on system costs and the feasibility of minimizing the visual impacts of renewable infrastructure under net-zero targets. In the present study, the lowest visibility distance is utilized, assuming that only major landscape changes from renewable energy are undesirable. Assuming a higher visibility distance further excludes land for renewable energy and would significantly increase the overall cost.”*

10. In terms of policy implications, the paper suggests that moderate visual restrictions do not heavily affect system costs. However, what specific policy recommendations can the authors offer for local governments balancing the need for renewable energy with visual and environmental concerns? Concrete examples would improve the paper's practical utility.

Thank you for your question. It is indeed an important point to give a concrete policy recommendation for local governments to balance the trade-offs of renewable energy demands with visual landscape concerns. We have now added the following sentences in the discussion section.

In line 330: “Furthermore, given the variability in population density and scenic areas across the country, a one-size-fits-all policy approach may be ineffective in implementing concrete policy at the local level. The results we present here show that there are some no-regret locations for renewable energy installations that are neither visible from scenic nor densely populated landscapes. Local governments could develop guidelines that vary according to local scenic and population characteristics to ensure that renewable energy development aligns with both energy and aesthetic priorities. For example, by prioritizing buffer zones based on viewsheds from scenic or population thresholds (e.g., scenicness = 9 and population density ≥ 3500 people/km²), policymakers can achieve a balance between preserving aesthetic value and promoting renewable deployment. In addition, policymakers could incentivize offshore wind or rooftop PV projects in visually sensitive areas through measures such as expedited permitting [73] or tax breaks [74].”

11. The reliance on hydrogen imports in high-sensitivity scenarios raises concerns about energy security. Could the authors explore alternative strategies to mitigate the reliance on imports, such as increased domestic production or enhanced storage technologies?

Thank you. To address these concerns of reliance on hydrogen imports at high-visual-sensitivity scenario, we could promote flexibility in the hydrogen supply chain for example through developing domestic production hubs from early on. However, in a high-visual-sensitivity scenario, renewable energy plants for green hydrogen production would only be available at locations with high levelized cost. Therefore, this strategy would likely add significantly to the overall system cost.

We now tried to explore other solutions by limiting the allowable hydrogen import ratio in additional optimization scenarios under high-visual-sensitivity. In these cases, the cost optimal strategy includes increasing building renovations to reduce electricity demand as measures to increase energy efficiency in the demand side. Moreover, the cost optimal strategy tries to tap on domestic biomass resources to compensate for not importing more hydrogen.

We added the following sentences in the discussion section.

In line 296: “Other strategies, such as promoting flexibility in hydrogen supply chains through the development of domestic production hubs, may reduce dependence on hydrogen imports. However, increased domestic production of green hydrogen would add

significantly to the overall system costs, as at this high visual sensitivity scenario, the few available renewable energy plants available for hydrogen production are located in high leveled cost locations. Tapping into domestic biomass potential or implementing demand-side management through building retrofits to reduce electricity demand are other more cost-effective strategies that could be explored to reduce the reliance on green hydrogen imports.”

12. The results indicate that excluding renewables from scenic and densely populated areas would lead to a more concentrated deployment in less visible regions. Has the paper considered the potential environmental justice issues, where the burdens of renewable energy infrastructure might disproportionately fall on less populated, rural areas?

Thank you for bringing up this important discussion point. We have briefly mentioned the potential concerns about distributive justice in the discussion section. When considering what is just, it is important to consider not only spatial distribution of renewable infrastructure, but also other tenets of justice, i.e., procedural and recognition, which are equally important (Vagero et al., 2024). Even when focusing on the spatial distributive justice, the distribution of renewable energy placement can vary widely depending on which principle is used to define and operationalize what is spatially just (e.g., equality, ability, benefit, etc.; Lehmann et al., 2024 and Vagero et al., 2024). For example, when focusing on the ability principle in terms of available land area, it might be more spatially just to place more wind turbines in rural areas that have more ability in hosting wind turbines compared to urban areas. Furthermore, it is also evident that the spatial distribution of local benefits and burdens depends not only on where renewable infrastructure is located, but also on how the benefits of renewable energy installations are redistributed (Lehmann et al., 2024). For example, through local ownership, access to cheaper renewable electricity, etc.

The real value of our reverse viewshed analysis is that it enables the integration of public concerns about the landscape impact of large-scale renewable energy early in the quantitative planning process. This has not previously been possible due to the limitations of conventional viewshed analyses. Our approach would promote public participation in the planning process and strongly support procedural justice. This is a first step in fostering participatory processes and, when applied at the local level, can be used to identify local preferences that may help realize a distribution of outcomes that is perceived as just by the local people.

We now added the following sentences in the discussion:

In line 305: *“High visual sensitivity from scenic or densely populated areas would also require renewable infrastructures to be sited mainly offshore or in certain remote areas, as shown in Fig. 2. This may concentrate environmental burdens in the locations not visible from scenic or populated areas and further give rise to concerns about distributive justice, given the potential for spatially unequal benefits and burdens associated with such strategies [68, 69]. However, it is important to note that the spatial distribution of local benefits and burdens depends not only on where renewable infrastructure is located, but also on how the benefits of renewable energy installations are redistributed [70]. For example, through local ownership structures and access to affordable renewable energy. To achieve a just transition, it is important to understand what people perceive as just. The reverse viewshed method demonstrated in this study can be applied at the local level to incorporate local preferences regarding the visibility of renewable infrastructure in the planning process. While the hidden placement of renewable energies demonstrated in this study may be a strategy from the spatial planners’ perspective, this approach should not preclude public participation in the decision-making process. Rather, it should be used as a means to reflect public preferences in the planning process.”*

Reviewer #2 (Remarks to the Author):

The main interest of this paper is that it presents, in a reasonably quantified manner, the cost of making wind and large scale PV energy compatible with the scenic value of a landscape. This is a highly topical issue, with public opinion controversies in almost every country. Since a large fraction of the interest, and merit, is the quantified analysis, an obvious shortcoming is that the results are valid for only one country, viz Germany. However, even if taken as an example, the solid quantification (the flaws reported below notwithstanding) make the paper a relevant, significant contribution to the debate on the visual impact of renewable energies.

Thank you for your valuable feedback. We have incorporated your suggestions into the revised manuscript. Please refer to our response in more detail below for each point.

The modelling work can be thought of as consisting of two parts: the viewshed model, which determines the visually-affected areas, and the energy-economic model, which determines the impact in the economy of different renewable-energy scenarios.

The viewshed model is sufficiently described and appears to be robust. The main weakness is that the model spatial resolution (1 km²) is arguably insufficient to capture many small-

scale features that would block the view, and hence perhaps allow a greater installed capacity in each scenario (particularly under high-population-density restrictions). This is briefly (and not very clearly) acknowledged by the authors in the Limitations section. But the way that this issue is addressed and discussed minimises the problem. The authors appear to argue (line 464 and following ones) that this effect is negligible because the difference between using a DTM and DSM is very small (0-0.5% of the eligible area). I would argue that this small difference is perhaps the proof that the relatively large resolution (of 1 km) is not adequate to capture this effect of small-scale features (which are under 1 km) blocking the view. While I acknowledge that the use of smaller resolutions may not be feasible for a study at the national scale, I think that the authors may be downplaying this effect, and perhaps they should clearly warn about it early in the paper.

Thank you for the comments. Unfortunately, we failed to clearly describe our methods in our original manuscript and ended up with an unclear explanation of our setup for the viewshed analyses. We have now made significant changes throughout the text to clarify that we used EU-DEM v1.1 DSM with 25-m spatial resolution for the basis of our reverse-viewshed analysis. The 25-m elevation resolution is high enough to capture small-scale features that may block the view. The centroids of 1 km² grid were only used as viewpoints. This 1-km raster was used to categorize areas for their scenicness and population density level to reflect landscape importance, but the viewshed analysis was implemented in high resolution, i.e., 25-m. In this regard, we have focused our revisions on this matter particularly on improving the description of our methodology. The comprehensive revision we carried out to better describe our methods regarding the different grid sizes utilized, are as follows:

1. Changes at the start of the results section, which we understood from the reviewer's comments, to be the major cause of this confusion. We now describe our methodology in a much clearer way from early in the text, particularly noting the DEM used for the analysis and its grid size (25 m) as well as the use of the 1 km centroids only as viewpoints:

In line 103: *"The reverse viewshed analyses were conducted on the entirety of Germany using the Copernicus EU-DEM v1.1 25 m grid [55]. The observer points were positioned at every centroid of a 1-kilometer-square grid that have underlying metadata of scenicness level and population density. In total, viewshed analyses were conducted from 357,588 viewpoints to determine theoretically visible areas from viewpoints with different scenicness and population density levels. The setup of the basic parameters of the viewshed analysis are presented in Fig. 1a."*

2. We made the respective changes to the start of the Methods section as well:

In line 384: *“Our methodology is divided into four steps (see Fig. 6). First, reverse viewshed maps were calculated on the EU-DEM v1.1 (with a raster size of 25 m [55]) for 357,588 viewpoints representing every kilometer-square of Germany.”*

In line 413: *“We performed the calculation of R-ZTV using the r.viewshed function in GRASS-GIS [85]. The inputs required for this analysis are a high-resolution Digital Elevation Model (DEM), viewpoints representing important landscapes, renewable energy infrastructure heights, observer height, and visibility thresholds. For DEM data, the Digital Surface Model (DSM) raster file for Germany from Copernicus EU-DEM v1.1 [55] with 25-m resolution was used. The viewpoints represent every centroid of each km² of Germany with underlying metadata of scenicness ratings [57] and the population data [86]. A total of 357,588 viewpoints representing the centroids of each km² of Germany was utilized.”*

3. We made auxiliary changes in Fig 1 and its caption.

“Fig. 1 Methodology of reverse visibility analysis and the use of the generated reverse viewshed maps as exclusion zones in land eligibility assessments. a., Reverse viewshed analyses are carried out to map the areas that, if selected for the installation of new wind turbines or open-field PV, would be visible from areas of high scenic quality or high population density (see Methods for more details). The viewshed analysis was conducted on the EU-DEM v1.1 with 25 m resolution. The viewpoints utilized were the centroids of a 1-kilometer-square grid of Germany with underlying metadata of scenicness level and population density...”

Additionally, the small differences of results using DTM and DSM may also be due to the slightly different resolution of the only available DTM raster, that is 20-m with the 25-m DSM raster. We added a sentence to further explain this in the limitation sector:

In line 623: *“A slight difference in grid size between the only available DTM raster (20 m) and DSM raster (25 m) may also contribute to this variation.”*

The energy system model is only loosely described in the paper. This is of course a very complex model, and a detailed description is perhaps impractical. The authors refer to three publications (refs 83, 84, 85) where the model "has been described, used, and extended". But, as with this paper, the description in those references is partial and insufficient, and very much tailored to the application in the respective paper. Given the relevance of the energy model in the present paper, an expanded description of the main hypotheses, features and methods of the model would be advisable. If space is a limitation, this could be submitted as supplementary material.

Thank you for your input. As you have noted, the model is highly complex and was developed over the course of numerous dissertations. We have now cited the dissertation by Peter Lopion (<https://publications.rwth-aachen.de/record/795478>) in the text, on which the most important equations and descriptions of the model are based. However, we must note that this dissertation is written in German, which is why we originally only cited the three studies we mentioned above. Furthermore, we have added the following information to the original description of the model in the method section:

In line 516: “.. The model enables the creation of normative scenarios for Germany’s future energy system and provides information on cost-effective ways to reduce greenhouse gas emissions. The model uses a linear optimization approach and is implemented bottom-up, i.e., the model maps individual technologies and components of the energy system in detail. Geographically, the model is limited to Germany but considers imports and exports of energy sources. For the present analysis, we chose a 5-year investment period, and the time horizon is set up to the year 2045.

The objective function is to minimize the total annual discounted system costs, including both fixed and variable costs. The fixed costs comprise capital expenditure and fixed operating expenses. The capital expenditure for a specific technology is determined by disaggregating the total investment into constant annuities, employing the capital recovery factor over the assumed depreciation period of that technology (which may differ from its technical lifetime). Additionally, the model considers the costs associated with supply infrastructure, such as electricity, heating and gas grids, as well as transport infrastructure.

In consideration of externally specified boundary conditions (e.g., greenhouse gas reduction targets) and assumptions (e.g., industrial goods production, transport demand), the most cost-effective combination of technologies and energy sources that simultaneously satisfies all constraints is determined. The transformation pathways are determined based on the cost-optimized hourly-resolved operating plans for all installed technologies. The cost-optimal pathways can be interpreted as the decision of a “central” planner and represent a macroeconomic perspective. The macroeconomic perspective ensures that the possible technologies or measures are not influenced by any external factors, as neither current tax nor subsidy mechanisms are considered. Consequently, the scenarios presented here represent cost-optimized energy transformation pathways that are not intended to forecast the future. Rather, they are designed to demonstrate what is theoretically possible. This approach ensures that the real cost-optimal transformation pathway is not biased by the potentially misplaced subsidies currently present in the German energy system.

The technologies that can be installed to meet the hourly energy demand and energy-related materials of all sectors are divided into generation, conversion and storage technologies. The model includes relevant future technologies and measures (e.g., PV, wind power, heat pumps, short- and long-term storage, retrofitting of buildings, etc.). The feed-in from energy sources is limited by their time-dependent potential and efficiency (e.g., weather-dependent feed-in profiles for PV- and wind power technologies). The installable capacities of the technologies have upper potential limits that are defined by technical restrictions [56] and in some cases (PV and wind power) also by the visibility restrictions in the scenarios considered in this study. Energy conversion in power plants and other facilities is determined by their efficiency and capacity. The operation of energy storage systems is subject to restrictions in terms of charging and discharging rates and storage capacity.”

The ease of interpretation of the methodology and results is at times marred by convoluted, vague or poorly thought-out presentation (which goes beyond a correct English usage). The authors should carefully review the manuscript so that the text is easy to understand on a first reading. Examples of instances where better language is needed are (partial list):

Thank you. We have carefully reviewed the manuscript and addressed your comments to improve the readability of the text. Please also refer to the changes that we made below.

- The caption of Fig 1, in particular subfigure b

Thank you. We have now improved the caption of Fig.1 as follows:

Fig. 1 Methodology of reverse visibility analysis and an exemplary use of the generated reverse viewshed maps as exclusion zones in land eligibility assessments. a., Reverse viewshed analyses are carried out to map the areas that, if selected for the installation of new wind turbines or open-field PV, would be visible from areas of high scenic quality or high population density (see Methods for more details). The viewshed analysis was conducted on the EU-DEM v1.1 with 25 m resolution. The viewpoints utilized were the centroids of a 1-kilometer-square grid of Germany with underlying metadata of scenicness level and population density. The visibility threshold distance was set at 11 km for wind turbines and 7.5 km for open-field PV. b., Example of integrating reverse viewshed maps from viewpoints with high scenicness level (level 9), and high population density (≥ 5000 people/km²) into the land eligibility assessment of onshore wind turbines in the district of Aachen, Germany. The green areas represent the eligible areas for siting of onshore wind turbines based on legal, geographical, technical, environmental, and additional visibility restrictions.

- The interpretation of Fig 2 (in particularly, the fact that the horizontal axes move through the several scenarios on top of the graph)

Thank you. The panel in the graph containing line plots shows the remaining renewable capacity potentials and the percentage of population that would get protected from the visual impacts of renewable installation at each visibility scenario. We have now revised Fig. 2 and improved the captions.

“Fig. 2 Reverse viewsheds for different scenicness and population density thresholds as exclusion zones and the remaining renewable energy potential. Capacity potential, shown as line graphs, are calculated after considering legal, geographical, technical, environmental, and additional reverse viewshed constraints. The secondary y-axis in the line plots shows the percentage of the population protected from visual impacts of renewable installations.”

- Supplementary figures 1 and 2 are in my opinion irrelevant for the goal of the paper
Thank you. We have now removed the Supplementary Figures 1 and 2.

- Figures like Supplementary Fig 3 would benefit from a legend, in each subfig, for ease of interpretation

Thank you. The supplementary figure (now became Supplementary Figure 1) has been updated to include legends for each subfigure. Additionally, the figure caption has been revised for ease of interpretation.

“Supplementary Figure 1: The visibility of existing onshore wind turbines (a) and open-field PV projects (b) from areas of high scenicness (scenicness level = 9) and high population density (population density ≥ 5000 people/km²). A mere 3% of existing onshore wind turbines and 2% of open-field PV installations are visible from the most scenic areas. Conversely, 12% of the existing onshore wind turbines and 2% of open-field PV installations are visible from densely populated areas. Similar analyses were also conducted for all other visibility threshold scenarios to analyze the visibility of the currently existing renewable infrastructures.”

- The meaning of placement density in Supplementary Fig 3 is not clear, and hence the conclusion ("The existing placement density is observed to decline with an increase in scenicness level") is hard to validate

Thank you. Using the metric of placement density, we analyze the visibility trends of existing onshore wind turbines and open-field PV projects in Germany. Placement density is calculated by dividing the number of visible wind turbines or PV projects by the visible area (in km²) derived from reverse-viewshed analysis for each important landscape category. These categories include:

- Reverse-viewshed areas originating from viewpoints with scenicness levels of 9, 8, 7, 6, 5, and <5.
- Reverse-viewshed areas from viewpoints categorized by population density: >5000, 3500–4999, 1500–3499, 300–1499, and <300 people/km².

This metric helps identify the concentration trends of wind turbines and PV projects based on their visibility across different landscape characteristics (e.g., highly scenic areas, moderately scenic areas, densely populated regions, etc.). Our findings indicate a higher concentration of wind turbines and PV projects visible only from areas with low scenicness levels (<5), while visibility from areas with scenicness ≥5 is significantly lower. This trend is illustrated by the descending line graphs in Supplementary Figure 2a. In conclusion, the visibility of existing renewable energy infrastructure declines with increasing scenicness, indicating a preference for installations in less scenic areas.

We have now revised the figure caption:

“Supplementary Figure 2: Concentration of the existing renewable energy plants by their visibility from different scenicness (a.) and population density levels (b.). The placement density (in line graphs) is calculated by dividing the number of existing wind turbine or open-field PV projects that are visible from different scenicness and population density levels by the reverse-viewshed areas in km² generated only from the respective scenicness or population density levels. In subfigure a, there is a relatively high concentration of onshore wind turbines and open-field PV that are visible only from areas with low scenicness levels (< 5), but not visible from areas with scenicness level ≥ 5. As the scenicness level increases, the concentration of these renewable energy plants decreases. For population density, this trend is only visible for open-field PV.”

- The meaning of “sectors” in Fig 4a, Supplementary Fig 5 and similar ones is unclear: it seems to heterogeneously mix economy sectors (eg Transport or Energy) with others that may be cost categories (eg, Conventional Fuels or Renewable Fuels, which are part of the energy sector)

Thank you. In Fig. 4a and other figures, we mix the visualization of cost deviation based on economic sectors (i.e., transport, industry, building, and energy) and other important sub-

categories of the energy sector (i.e., grid infrastructure, storage, imported renewable fuels, and imported conventional fuels) for the ease of visualization of the trends of these important sub-categories. However, we failed to explain it clearly in the figure caption. We have now revised the figure caption as follows:

In Fig 4: *Fig. 4 Relative cost deviation by category (a) and relative deviation of electricity supply by source (b) when large-scale renewable energy plants are not visible from scenicness levels ≥ 5 and population density ≥ 300 people/km². a. The cost deviation is shown for each economic sector and important energy-related sub-sectors. The energy sector accounts for domestic energy supply. The infrastructure sector accounts for grid costs. Renewable and conventional fuels represent the cost for imported fuels. Other sectors, such as the building stock or transport options, are not as strongly affected as the energy sector. At the strictest visibility restriction, the cost from energy sector increases by 38% in 2045 compared to the base scenario (equivalent to €23.6 billion). b. Reduction in electricity supplies from onshore wind and open-field PV are substituted by rooftop PV and offshore wind. The blue line shows the need to increase hydrogen imports shares in fulfilling demand at the strictest visibility restrictions.*

In Supplementary Fig.3: *“Overall system cost by sector and major energy-related sub-sectors across visibility scenarios.”*

In Supplementary Fig. 4: *“Relative deviation of overall system cost by sector and major energy-related sub-sectors across scenarios.”*

- The caveat in line 442 and following is vague and poorly expressed.

Thank you. We have now deleted this paragraph and integrated it to line 538:

“...The cost-optimal pathways can be interpreted as the decision of a “central” planner and represent a macroeconomic perspective. The macroeconomic perspective ensures that the possible technologies or measures are not influenced by any external factors, as neither current tax nor subsidy mechanisms are considered. Consequently, the scenarios presented here represent cost-optimized energy transformation pathways that are not intended to forecast the future. Rather, they are designed to demonstrate what is theoretically possible. This approach ensures that the real cost-optimal transformation pathway is not biased by the potentially misplaced subsidies currently present in the German energy system.”

The authors need to critically examine the readability of the paper and edit it with clarity in mind.

Thank you. We have carefully reviewed the manuscript and addressed your comments to improve the readability of the text. Please refer to the changes that we made.

Under the section "Data availability", the authors provide links to public repositories with codes and tools. The authors refer to the information provided as "frameworks", "code", "software tools". Inspection of the links provided appears to indicate that in some cases the software and data provided would allow to reproduce the paper results (this is perhaps the case of the reverse viewshed analysis), but this is not so clear in other cases. For instance, the tool for land eligibility analysis appears to apply only to Aachen; and in the case of ETHOS.FINE it would appear that the software link is for the general framework, rather than for its instantiation for this work (eg with relevant data).

Thank you. The code and data for reverse viewshed analysis stored in the reference link indeed allow for reproducibility. For the land eligibility and placement model, renewable energy simulation model, and energy system model for Germany, we refer to the open-source frameworks used in this study (ETHOS.GLAES, ETHOS.RESKit, and ETHOS.FINE respectively). While most of the inputs to these models are openly available (such as the used distance restrictions or used OpenStreetMap data), some data to build the models are not publicly available due to data protection reasons. For example, the partial use of data under data protection law was necessary in order to assess the potential of renewable energies at an unprecedented level of detail (see the publication by Risch et al. 2022).

We revised the Code and Data availability statement as follows:

“All models used for this study are open source. The developed code for the large-scale reverse viewshed analysis can be accessed on Jülich DATA, doi:10.26165/JUELICH-DATA/PLNH9P. The software tool ETHOS.GLAES, used for land eligibility analysis can be found at <https://github.com/FZJ-IEK3-VSA/glaes>. The ETHOS.RESKit tool, which is employed for the electricity generation simulation can be accessed via <https://github.com/FZJ-IEK3-VSA/RESKit>. The data that support the findings of this study are partially publicly available (e.g., land use data from OpenStreetMap or distance restrictions in Supplementary Table 1). However, the use of some data is restricted, as for example the high-resolution building data from the German Federal Agency for Cartography and Geodesy (BKG). Finally, the ETHOS.Fine energy system optimization framework can be found at <https://github.com/FZJ-IEK3-VSA/FINE>. In contrast to the above models, this latter reference leads only to the optimization framework, which can be used to instantiate a model such as the one of the national energy system used in this study. We are currently also working on an open-source publication of this model. In the meantime, please contact the authors for more information on the model setup and, if not included in this manuscript, the data.”

Lastly, while English usage should not be addressed in this review (and it is not), because of its pervasiveness in the paper, I need to stress that scenicness is not, I think, an English word, and it is an ugly invention. Perhaps the authors can consider alternatives, such as scenic value or scenic quality?

Thank you for your suggestion. You are right, the word scenicness is indeed not an official English word. However, in the fields of spatial planning, geography, health and environmental well-being, and economics (for stated preference studies), the word scenicness has been used in scientific discourse since the 1960s in several publications^{1,2}. Scenicness refers to the perceived aesthetic and scenic beauty of observed landscapes. More recently, the word has been frequently used in scientific articles in the fields of geoinformatics and remote sensing³⁻⁸, environmental well-being⁹, and energy system planning¹⁰⁻¹⁴. The references cited are only a few examples. We believe that due to the relevance of the cited articles to our study, it is important to maintain the use of the word scenicness.

1. Board, N. R. C. (U.S.) H. R. *Highway Research Record*. (National Research Council, Highway Research Board, 1963).
2. Davidson, P., Tomer, J., Waldman, A. & Research, R. U. B. of E. *The Economic Benefits Accruing from the Scenic Enhancement of Highways*. (Rutgers, the State University, 1968).
3. Arendsen, P., Marcos, D. & Tuia, D. Concept Discovery for The Interpretation of Landscape Scenicness. *Machine Learning and Knowledge Extraction* **2**, 397–413 (2020).
4. Chang Chien, Y.-M., Carver, S. & Comber, A. Using geographically weighted models to explore how crowdsourced landscape perceptions relate to landscape physical characteristics. *Landscape and Urban Planning* **203**, 103904 (2020).
5. Levering, A., Marcos, D. & Tuia, D. On the relation between landscape beauty and land cover: A case study in the U.K. at Sentinel-2 resolution with interpretable AI. *ISPRS Journal of Photogrammetry and Remote Sensing* **177**, 194–203 (2021).
6. Levering, A., Marcos, D., Jacobs, N. & Tuia, D. Prompt-guided and multimodal landscape scenicness assessments with vision-language models. *PLOS ONE* **19**, e0307083 (2024).
7. Havinga, I., Marcos, D., Bogaart, P. W., Hein, L. & Tuia, D. Social media and deep learning capture the aesthetic quality of the landscape. *Sci Rep* **11**, 20000 (2021).
8. Jeawak, S. S., Jones, C. B. & Schockaert, S. Predicting the environment from social media: A collective classification approach. *Computers, Environment and Urban Systems* **82**, 101487 (2020).
9. Seresinhe, C. I., Preis, T. & Moat, H. S. Quantifying the Impact of Scenic Environments on Health. *Sci Rep* **5**, 16899 (2015).
10. McKenna, R. *et al.* Scenicness assessment of onshore wind sites with geotagged photographs and impacts on approval and cost-efficiency. *Nature Energy* **6**, 663–672 (2021).
11. Weinand, J. M. *et al.* Exploring the trilemma of cost-efficiency, landscape impact and regional equality in onshore wind expansion planning. *Advances in Applied Energy* **7**, 100102 (2022).
12. Weinand, J. M., McKenna, R., Kleinebrahm, M., Scheller, F. & Fichtner, W. The impact of public acceptance on cost efficiency and environmental sustainability in decentralized energy systems. *Patterns* **2**, 100301 (2021).
13. Price, J., Mainzer, K., Petrović, S., Zeyringer, M. & McKenna, R. The Implications of Landscape Visual Impact on Future Highly Renewable Power Systems: A Case Study for Great Britain. *IEEE Transactions on Power Systems* **37**, 3311–3320 (2022).
14. Lohr, C. *et al.* Multi-criteria energy system analysis of onshore wind power distribution in climate-neutral Germany. *Energy Reports* **12**, 1905–1920 (2024).

Reviewer #3 (Remarks to the Author):

A very interesting, timely and relevant manuscript, which addresses a problem that has not received sufficient attention: the cumulative, regional, potential impact by visibility and its influence on pathways of the energy system transformation. A few aspects of the methods applied, however, remain opaque in the present manuscript.

Thank you. In response to your comments, we have revised the manuscript with the aim to improve the explanation of our study setup.

General discussion:

Visibility is not equal to visual impact, i.e. a location exposed to visibility is not necessarily perceived as an unsuitable location. I believe that Shang and Bishop (2000) have pioneered the distinction between detection, recognition and impact in their paper “Visual thresholds for detection, recognition and visual impact in landscape settings. However, while there is little empirical evidence on the degree of visual impact for cumulative turbines or population, the question is whether visual impact is really a problem for the residents of population centres or the visitors to scenic places, or is it impossible to relate the intensity of felt visual impact to the preferences of people, like Ladenburg (2015) discussed in “Does more wind energy influence the choice of location for wind power development? Assessing the cumulative effects of daily wind turbine encounters in Denmark”? Also, is local opposition not usually a matter of a small minority residents? I believe the paper needs a more thorough discussion of this aspect.

Thank you for raising this important point of discussion. We agree with the considerations raised and have incorporated them in the setup of our study but were not successful in demonstrating this in the writing of our manuscript. We added the following in our revised manuscript:

1. In the method section:

In line 442: “Secondly, it is acknowledged that visual impact is inherently a matter of subjective landscape perception, thus it embodies important uncertainties. These uncertainties refer to both current differences over perceptions [26, 30, 75, 76, 90, 93] and also to the potential for positive-negative shifts in perceptions over time [13, 77, 94, 95]. Consequently, although calculations of visual impacts are valuable, their integration in planning should be carefully considered and responsive to evolving social preferences over time. R-ZTV is an effective approach in this regard, as the generated maps can be utilized in the future both as exclusion zones or as weighted spatial layers in multi-criteria studies that consider local preferences at smaller planning scale.”

2. In the discussion section:

In line 359: “While the scenarios presented here are based on empirical studies that indicate that the dominant visibility of renewable energy on scenic landscapes is not preferred and leads to rejection [24–29], it is important to note that visibility of renewable energy infrastructure is not always universally perceived as a negative visual impact by the public. For instance, in the case of wind energy infrastructure, perceptions of wind turbines can range from fully positive ones originating from feelings of progress and sustainability to fully negative ones induced by critique of landscape industrialization [30, 75]. Moreover, even negative perceptions are not always a direct indication of actual willingness to oppose projects, since it has been demonstrated that opposition to projects is often led by dedicated ‘vocal minorities’ [13, 26, 76], other than by institutional means such as from administrative or voted local authorities [30].”

3. In the limitations section:

In line 668: “Furthermore, while the investigation of visibility from scenic and densely populated areas partially quantifies the visual impacts of infrastructure on important landscapes [42], not all aspects of visual impacts [51] are included in the present analysis. The concept of visual impacts also depends on the level of detection, recognition [111], perceptions of annoyance from renewable energy infrastructure, and is also influenced by place attachment [112]. Over the past decade, it has also become evident that the perception of visual impact from renewable energy infrastructure is also not a strictly independent criterion for individuals, as it is intertwined with other parameters that affect local acceptance. Such factors may include community participation in the design and planning process [77], the application of landscape studies in the planning process [94], the pre-existing character of the landscape [95, 96], or even professional backgrounds of the affected populations [13]. The acceptance rate may also change over time [113] and be influenced by local people’s familiarity and experience with renewable energy projects [114], the dynamic changes in people’s sensitivity to visual impacts from renewable infrastructure may also be a source of uncertainty for landscape impact quantification. This issue would be complex to model, as in our current study, the renewable energy plants at the various locations are only input data prior to optimization and it is not clear which sites would be selected in which investment period based on cost optimality or other criteria. To reflect this, it would be conceivable in future studies to develop an iterative approach that assesses the optimization results considering the evolving societal preferences depending on the number of installed renewable plants in a given region and then adjusts the input data. Additionally, site-specific visual impacts, such as shadow flicker phenomena from wind turbines [115], glare from PV [116], or varying severity of visual impacts due to

cumulative number of concentrated renewable energy plants [75] could be incorporated in future analyses.”

A grid resolution of 1km may result in a significant error for line-of-sight analysis especially in hilly terrain, and also the site eligibility and LCOE calculations will have significant inaccuracy levels at this scale. The manuscript owes a discussion of the likely error induced by this coarse resolution. Data of at least 100m resolution are available, and computational effort should be manageable as well.

Thank you for your comment. We apologize for the unclear explanation of our setup in the previous version of the manuscript. We used the native EU-DEM v1.1 with 25-m spatial resolution [ref 55.] as the basis for our reverse-viewshed analysis. The 1-km grid is only used to sample viewpoint from the entire Germany. This 1 km raster was used to categorize areas for their scenicness, but the viewshed analysis from these viewpoints was implemented in high resolution, i.e., 25-m raster. The subsequent land eligibility analysis was also conducted according to Risch et. al., 2022 that utilizes 10-m resolution. Stanley Risch and Rachel Maier are also involved as the co-authors of this study. In this regard, we have focused our revisions on this matter particularly on improving the description of our methodology. The comprehensive revision we carried out to better describe our methods regarding the different grid sizes utilized, as follows:

In line 384: *“Our methodology is divided into four steps (see Fig. 6). First, reverse viewshed maps were calculated on the EU-DEM v1.1 (with a raster size of 25 m [55]) for all 357,588 viewpoints representing every kilometer-square of Germany.”*

In line 414: *“The inputs required for this analysis are a high-resolution Digital Elevation Model (DEM), viewpoints representing important landscapes, renewable energy infrastructure heights, observer height, and visibility thresholds. For DEM data, the Digital Surface Model (DSM) raster file for Germany from Copernicus EU-DEM v1.1 [55] with 25-m resolution was used. The viewpoints are every centroid of each km² of Germany with underlying metadata of scenicness ratings [57] and population data [86]. A total of 357,588 viewpoints representing the centroids of each km² of Germany was utilized.”*

In line 482: *“ETHOS.GLAES allows binary land exclusions in the study region based on user-defined criteria. The base scenario employs multiple high-resolution (as fine as 10-m) land exclusion maps including legal, geographical, technical, environmental, and cultural preservation restrictions (see Supplementary Table 1) as in the potential analyses conducted by Risch et al. [56].”*

The results are presented in a meaningful way, and valuable conclusions are drawn. However, the main parameters such as the cut-off distance and the definition of what is scenic beauty and dense population are fixed without exploring any sensitivity that may be contextual to the problem of the acceptance of wind energy and the role that visibility plays in it. E.g. most literature agrees that ownership form, scale of installation, or procedural justice in planning wind farms has an equally high if not higher importance.

Thank you for suggesting this important point of discussion. We have carefully selected the parameters and scenarios for our analysis; however, we failed to demonstrate it clearly in the previous version of the manuscript. We now added discussion on cut-off visibility distances and scenario selection in the method section.

In line 431: *“The maximum distance of visibility, also referred to as the visibility threshold [89], exerts the most significant influence on the results of the R-ZTV analysis, as it defines the radius of application of the viewshed analysis [51]. In visibility analyses for large spatial scales, the utilized thresholds range from 10 to 35 km [30, 90]. When limiting the visibility effect to dominant visibility within a landscape, the range of thresholds could be reduced to 2 to 8.1 km [91, 92]. In our analysis we utilized a visual threshold of 10 km, and extended it by an additional 1 km, reaching 11 km. This adjustment accounts for the additional distance from grid border to the viewpoint in the center of each km² of the scenicness grid. The lowest edge of the spectrum of visibility thresholds is selected for two reasons: First, as the R-ZTV maps generated in our analysis are proposed as exclusion zones, a stricter definition of visibility is deemed appropriate. Secondly, it is acknowledged that visual impact is inherently a matter of subjective landscape perception, thus it embodies important uncertainties. These uncertainties refer to both current differences over perceptions [26, 30, 75, 76, 90, 93] and also to the potential for positive-negative shifts in perceptions over time [13, 77, 94, 95]. Consequently, although calculations of visual impacts are valuable, their integration in planning should be carefully considered and responsive to evolving social preferences over time. R-ZTV is an effective approach in this regard, as the generated maps can be utilized in the future both as exclusion zones or as weighted spatial layers in multi-criteria studies that consider local preferences at smaller planning scale. For solar PV, the selection of a visibility threshold was more straight forward, as there are only few studies that particularly referred to visual impacts from solar energy. Therefore, the radius used in the study by Palmer [96] was adopted: 6.4 km, extended by 1 km and rounded to 7.5 km, accounting for the additional distance to the viewpoint in the center of each km² of the scenicness grid.”*

And additionally in discussion section:

In line 359: *“While the scenarios presented here are based on empirical studies that indicate that the dominant visibility of renewable energy on scenic landscapes is not preferred and leads to rejection [24–29], it is important to note that visibility of renewable energy infrastructure is not always universally perceived as a negative visual impact by the public. For instance, in the case of wind energy infrastructure, perceptions of wind turbines can range from fully positive ones originating from feelings of progress and sustainability to fully negative ones induced by critique of landscape industrialization [30, 75]. Moreover, even negative perceptions are not always a direct indication of actual willingness to oppose projects, since it has been demonstrated that opposition to projects is often led by dedicated ‘vocal minorities’ [13, 26, 76], other than by institutional means such as from administrative or voted local authorities [30]. In addition, the findings presented here illustrate that reducing the visibility of renewable infrastructures to the greatest extent alone would be an expensive strategy for increasing acceptance. This underscores the significance of combining the visibility consideration in planning with other efforts to enhance local acceptance of renewable energy projects, such as by ensuring local involvement in the planning process [15, 77] and providing ownership schemes for local communities [78].”*

Specific comments

Introduction:

Chapter heading „Introduction“ is missing

Thank you. We have now added the Introduction chapter heading.

L 38-39: Rather than “mitigating the intensification of climate-related extreme events” I would say “mitigating climate change” in general. Besides extreme events, climate change also drives a general degradation of the climate.

Thank you. We have now revised the first sentence:

A key component in mitigating climate change is the substitution of conventional fossil fuels with sustainable, low-carbon, renewable energy [1].

L 43-45: Social opposition due to lack of acceptance is perhaps the most important, but not the only reason for reduced growth of on-shore wind energy in Germany. Others are political issues on the state level, supply chain problems of the industry, and perhaps most importantly delays in the planning and permitting of new wind farms, but also road infrastructure etc. Please consider to revise the motivation accordingly.

Thank you. We agree that while social acceptance is often cited as a significant factor in the declining growth of onshore wind in Germany in recent years, there are also bottlenecks due to lengthy permitting processes, supply chain issues, and limited grid expansion. We have now revised the motivation in the first paragraph of the introduction as follow:

In line 43: “In Europe, and particularly in Germany, after years of record capacity expansion, the growth rates have recently been declining sharply [8–10]. The decline has been attributed to a number of factors, including the lengthy permitting process [9, 11], supply chain issues, and insufficient grid expansion [12]. Above all, local opposition to renewable energy technologies, particularly wind power, has been identified as one of the most significant barriers to their deployment [3, 8, 13–15].”

L 213 ff: Please explain why rooftop PV requires more storage than in the base case. I believe the reason is a lower deployment of wind energy, but that should be made explicit.

Thank you. You’re right, in this scenario there is a much lower deployment of large-scale wind energy, that would make the storage requirements higher. We have now revised the sentence to make it explicit.

In line 229: “The massive deployment of distributed rooftop PV, accompanied by a notable decline in wind energy deployment necessitates huge storage development. This is reflected in the increases in storage sector costs by up to threefold at scenicness levels ≥ 5 and population density ≥ 300 people/km², compared to the base scenario.”

Methods:

The reverse use of the viewshed method sounds intriguing. However, how do you deal with cumulative viewsheds, which in standard ZVI analysis is a measure of the quantity of likely visual impact, as it counts the number of viewpoints visible at each location? Does the visibility of a single wind turbine have the same weight as the cumulative impact by a large number of wind farms?

Thank you. Since our viewshed analysis is carried out from perspective of important landscapes that are to be protected, we quantified potential visual impact of one wind turbine to multiple important landscapes. In the land eligibility model, as we used the combined reverse viewsheds as exclusion zone, it would place any number of wind turbines (i.e., one or large number) out of sight of multiple important landscapes. The cumulative viewshed analysis mentioned by the reviewer under the usual cases, can only be done after knowing the exact locations of planned wind turbines or wind farm. While in our setup, we focus more on finding suitable locations considering visibility from multiple important landscapes.

The question that remains is whether the addition of one or fifty turbines, visible from multiple scenic points, would have the same effect. If in the future study, cumulative reverse viewshed maps are utilized for the detailed design of a wind energy project, this could be considered by counting the number of planned wind turbines within the combined reverse-zone of theoretical visibility (R-ZTV) and weighing each turbine by the number of cumulative scenic points it affects. We now added this to the limitation of our study.

In line 642: “In our reverse viewshed analysis, we quantified the potential visual impact of a single wind turbine on multiple important landscapes and used it as an exclusion zone in the land eligibility model. This would result in placement of any number of wind turbines out of sight of multiple important landscapes considered. A limitation of the current settings is that it is not feasible to assess the land suitability based on the potential visual impact of single or cumulative numbers of wind turbines. This is because the necessary information regarding turbine locations and numbers must be known in advance of the energy system optimization and would be more appropriate for a smaller-scale analysis. Future studies for the detailed design of a wind energy project, could employ cumulative reverse viewshed analysis and count the number of planned wind turbines within the combined reverse-zone of theoretical visibility (R-ZTV). Each turbine could then be weighted by the number of cumulative viewpoints it affects.”

Besides errors in elevation grids, here partially considered by a comparison of DSM and DTM, the most significant source of error in viewshed analyses is the raster resolution of the DEM used (see e.g. Riggs and Dean, 2007). Please describe the method by which the native 30m (EU-LAEA) resolution of the EU-DEM v1.1 elevation model (which btw. is no longer supported via the link provided [76]) has been aggregated to 1km. For viewshed analysis, one could argue for particular requirements to resampling, as averaging may not be the most suitable. In any case, errors are induced, and the authors owe an account of the likelihood of errors induced by resampling the central data source.

Thank you. We apologize for the incomplete information in the previous version of the paper, which led to misunderstandings about the spatial resolution used in our reverse viewshed analysis. We would like to clarify that we did not aggregate the native EU-DEM v1.1 dataset. We directly used it and the spatial resolution of our reverse viewshed is 25 m. The 1-km grid is only used to sample the viewpoints for the reverse viewshed analysis. Thank you for pointing out that the link to the EU-DEM v1.1 dataset is unfortunately not available anymore, but the details about the dataset can be checked here:

<https://sdi.eea.europa.eu/catalogue/geoss/api/records/3473589f-0854-4601-919e-2e7dd172ff50> or here <https://www.eea.europa.eu/en/datahub/datahubitem-view/d08852bc-7b5f-4835-a776-08362e2fbf4b#tab-metadata>

We have now carried out a complete overhaul of the manuscript in relation to describing our methods. Including:

1. Changes at the start of the results section, which we understood from the reviewer's comments, to be the major cause of this confusion. We now describe our methodology in a much clearer way from early in the text, particularly noting the DEM used for the analysis and its grid size (25 m) as well as the use of the 1 km centroids only as viewpoints:

In line 103: "The reverse viewshed analyses were conducted on the entirety of Germany using the Copernicus EU-DEM v1.1 25 m grid [55]. The observer points were positioned at every centroid of a 1-kilometer-square grid that have underlying metadata of scenicness level and population density. In total, viewshed analyses were conducted from 357,588 viewpoints to determine theoretically visible areas from viewpoints with different scenicness and population density levels. The setup of the basic parameters of the viewshed analysis are presented in Fig. 1a."

2. We made the respective changes to the start of the Methods section as well:

In line 384: "Our methodology is divided into four steps (see Fig. 6). First, reverse viewshed maps were calculated on the EU-DEM v1.1 (with a raster size of 25 m [55]) for 357,588 viewpoints representing every kilometer-square of Germany."

In line 413: "We performed the calculation of R-ZTV using the r.viewshed function in GRASS-GIS [85]. The inputs required for this analysis are a high-resolution Digital Elevation Model (DEM), viewpoints representing important landscapes, renewable energy infrastructure heights, observer height, and visibility thresholds. For DEM data, the Digital Surface Model (DSM) raster file for Germany from Copernicus EU-DEM v1.1 [55] with 25-m resolution was used. The viewpoints represent every centroid of each km² of Germany with underlying metadata of scenicness ratings [57] and the population data [86]. A total of 357,588 viewpoints representing the centroids of each km² of Germany was utilized."

3. We made auxiliary changes in Fig 1 and its caption.

"Fig. 1 Methodology of reverse visibility analysis and the use of the generated reverse viewshed maps as exclusion zones in land eligibility assessments. a., Reverse viewshed analyses are carried out to map the areas that, if selected for the installation of new wind turbines or open-field PVs, those would be visible from areas of high scenic quality or high population density (see Methods for more details). The viewshed analysis was conducted on the EU-DEM v1.1 with a 25 m resolution. The viewpoints utilized were the centroids of a 1-kilometer-square grid of Germany with underlying metadata of scenicness level and population density."

4. We downloaded the EU-DEM v1.1 dataset over two years ago, and unfortunately it is not available anymore from the link <https://land.copernicus.eu/imagery-in-situ/eu-dem/eu-dem-v1.1>. We now changed the reference to show the metadata of the EU-DEM v1.1 dataset.

The choice of a 1 km raster resolution for land eligibility seems to have been motivated by the high computational effort required, and perhaps also by the scenic value grid used. However, land use maps are available at a 10 m resolution, as Risch, Mayer et al. (2022) suggest; and since Germany is characterised, in many places, by a highly fragmented land use pattern, see Schug, F. et al. (2020), the choice of a resolution a factor 10,000 less would induce a significant error. Would it be possible to estimate this error? On the other hand, the land requirement of a single reference turbine is roughly 1 km², so is the raster resolution chosen out of convenience for the analysis?

Thank you. This is also related with your previous comment, and we would like again to acknowledge that our description was not clear and that we have carried out extensive improvements to address this (including a more explicit description in the results section, in Fig 1 and in the methodology section). We would like to clarify that our land eligibility analysis used the same resolution (10 m) as Risch, Maier et al. (2022). Mr. Risch and Ms. Mayer are also the co-authors of this study.

I would argue that the scenic value dataset by Roth et al. (2018) could be interpolated to higher resolutions without loss of validity, thereby increasing the chance to better accommodate the complexity of landscapes.

Thank you for your suggestion. As previously clarified, the 1-kilometer-square scenic value data is only used as viewpoints. If the purpose of interpolating the scenic value dataset is to increase the resolution of reverse-viewshed results, then it should have been clarified with the previous point that we utilized the 25-m DSM grid to produce the reverse viewshed results that covered complex landscape elements at high resolution. We added this explanation in the method section. For more detailed analysis in a smaller spatial scale, it might be possible to interpolate the scenicness dataset to increase the number of viewpoints. We have now revised the limitation section regarding this matter:

In line 600: “Secondly, the scenicness dataset employed as viewpoints in the reverse viewshed analysis has a relatively low resolution of 1 km. This may result in the omission of heterogeneity in landscape quality within a 1 km² area. Nevertheless, it should be noted that the centroids of the 1 km² grid are employed solely for the purpose of sampling viewpoints across Germany with diverse landscape quality. The impact on the resulting reverse viewshed map is anticipated to be minimal. This is due to the fact that the resulting

viewshed map is largely contingent upon the digital elevation map employed and the visibility thresholds used. The present study used the EU-DEM v1.1, which is capable of capturing small-scale features at 25 m resolution. Additionally, as stated above, the assumed visibility thresholds have been adjusted to account for the additional distance to the viewpoint in the center of the grid. Future studies that aim to conduct a more detailed analysis at a smaller spatial scale could benefit from interpolating the scenicness dataset to increase the number of viewpoints.”

Census data (77): which data have been used? The native 100m grid must have been resampled; please describe how. Like for the other data layers, by resampling to 1 km² cell size the analysis sweeps over the complex texture of land use and settlement patterns in Germany. A discussion of the likely errors would be required.

Thank you. We used the official data of population in 1-kilometer grid from Zensus 2011 and did not do any resampling process.

https://www.opengeodata.nrw.de/produkte/bevoelkerung/zensus2011/ergebnisse_1km-gitter/. As we have clarified in our response to the previous points, the datasets that are in 1-kilometer-square resolution comprise solely the data on scenicness level and population density utilized as viewpoints in the reverse-viewshed model. The other datasets, namely the digital surface model (EU DEM v1.1), serve as the foundation for the reverse viewshed analysis and are in high resolution of 25-m. The land use dataset employed in the land eligibility model is in 10-m resolution.

Observer height and dealing with obstacles in land- and cityscapes: a uniform observer height is difficult to include, since especially in urban environments; people rarely see much of their surroundings (surrounded by buildings) while in rural settings, vegetation will obscure their view. Also, the scenic value map could assist here as there is (visually assessed) a high spatial correlation between high population density and low scenic beauty.

Thank you for raising this point. As we utilized the 25-m Copernicus EU-DEM v1.1 that represents the surface of the Earth including buildings, infrastructure, and vegetation, the point of observation in our current setting is assumed to be on top of the features, e.g., the viewshed from people standing on the rooftop terrace. Our result would show a conservative scenario in the urban settings, as the elevation is averaged out at 25 m resolution. When viewed from the lower elevation, i.e., the ground, the viewshed area at urban settings could be lower than the obtained results. However, to get into this detail level of analysis would require a detailed 3D city model that does not fit for our large-scale analysis.

We revised the discussion on this point at the limitation section:

In line 613: “Thirdly, the input of the reverse viewshed analysis employed in this study is a digital surface model (DSM), which captures both natural and artificial features, e.g., buildings and trees. At locations with a high number of features such as forests and cities, the reverse viewshed obtained might be overestimated since the observer point is assumed to be on top of these features. When viewed from the ground, the resulting viewshed area at urban settings could be lower than the results obtained from the present analysis. However, due to the large geographic scale of this study, the deviation is negligible. A sensitivity analysis (see Supplementary Fig. 14) indicates that the calculated eligible area exhibits minimal variation (0 – 0.5%) when either digital surface model (DSM) or digital terrain model (DTM) [109] is employed in the reverse viewshed calculation. A slight difference in grid size between the only available DTM raster (20 m) and DSM raster (25 m) may also contribute to this variation. A detailed viewshed analysis with much higher spatial resolution could combine the DSM with a digital terrain model (DTM), which is a bare-earth elevation model, to detect possible features blocking the line of sight. This improvement could be beneficial for a micro-level (i.e., municipal) analysis where exact locations of power plants need to be determined.”

L 345: While standard open field PV has a height of about 3 m, agri-PV would at least be 4 m high, depending on the type. With an observer height of 2 m, the horizon distance is 5 km, hence Earth curvature becomes relevant.

Thank you for your comment. This is true, the height of 2 m is not representative of agri-PV and our explanation in this matter was inadequate. Also, you are correct that at 5 km (or 7.5 km PV visibility distance in our case), there might be an effect of earth curvature on the visibility of 2 m-PV. We have now explained our choices in a better way in this regard and clarified the selection of heights to avoid confusion. Firstly, the reason for deciding to omit the earth’s curvature from the analysis is that it was considered a simple way to improve the performance of our algorithm that had to carry out viewshed analysis for 357,588 viewpoints on 25 m elevation data. Secondly, to compensate for potential overestimation of reverse viewshed area for PV calculation, we chose the lower heights of PV panels.

We revised our statement in the method section in line 455: *“Furthermore, additional considerations such as earth curvature and atmospheric refraction were omitted from the analysis as it requires more computational power. The exclusion of earth curvature consideration from the analysis was an additional rationale for selecting a lower end height for solar PV. This was done to counterbalance the potential for overestimation. For this reason, a height of 2 m was chosen for open-field PV panels, which range in height from 2 to 5 m in agrivoltaics applications [97, 98]”.*

In addition, we performed a sensitivity analysis for a medium-sized region to find out the difference in reverse-viewshed area for PV when Earth curvature and atmospheric refraction are considered. We found that our results without considering Earth curvature and atmospheric refraction cause 1-2% larger reverse-viewshed areas. We added this to the limitations of our method.

In line 637: *“The reverse viewshed analysis employed in this study did not consider the effects of earth curvature and atmospheric refraction. This resulted in an overestimation of the visible area for 2 m solar PV by 1–2%. This effect is lower for larger objects such as onshore wind turbines. Future studies could consider earth curvature and atmospheric refraction in the viewshed analysis to improve accuracy.”*

L 358: it is not clear if boundary effects are included (by considering viewpoints up to 11 km outside the boundaries of counties to include the visual impact from turbines located in neighbour counties) and whether the inclusion of the boundaries would not lead to an unnecessary high computational effort because the total number of line-of-sight analyses increases significantly.

Thank you for sharing this point for consideration. We have improved our phrasing in the manuscript to better describe this aspect of our work. Since our application was carried out for the total area of Germany, all counties and regions were included in the analysis. We parallelized the analysis on our high-performance computing cluster by each county to speed up the process. In the end, we combined the visibility map from each county. Therefore, it was important to accurately calculate viewshed areas, even when the viewpoint was close to a county’s boundary that would extend its viewshed area up to 11 km to the neighboring county. In our approach, for each county we used up to 11 km visibility distance to accurately calculate the viewshed from viewpoints that are located at county’s boundary. In these cases, the digital elevation model was also considered beyond the county boundaries. In Fig. 1b, this is visible for the Aachen district with all viewpoints contained inside the county’s boundary. We have in total 357,588 viewpoints, representing every centroid of 1-kilometer-grid of Germany.

We acknowledge that in our initial description this was not very clear. We made the following three improvements: (i) We improved the starting paragraph of the results section, which now makes all those study design decisions clear from early on in the manuscript:

In line 103: *“The reverse viewshed analyses were conducted on the entirety of Germany using the Copernicus EU-DEM v1.1 25 m grid. The observer points were positioned at every centroid of a 1-kilometer-square grid that have underlying metadata of scenicness level and population density. In total, viewshed analyses were conducted from 357,588 viewpoints to*

determine theoretically visible areas from viewpoints with different sceniness and population density levels.”

(ii) we added some descriptive text in Fig 1.:

a. Reverse viewshed analysis - parameter setup

b. Example of integrating reverse viewshed maps into land eligibility assesment for onshore wind placement on District of Aachen, Germany

and (iii) we improved the beginning of the Methods section:

“Our methodology is divided into four steps (see Fig. 6). First, reverse viewshed maps were calculated on the EU-DEM v1.1 (with a raster size of 25 m) for 357,588 viewpoints representing every kilometer-square of Germany.”

It is not clear why this process was computation intensive. I found it to be no problem to run the viewshed tools in QGIS and ArcGIS Spatial Analyst for a 100 m elevation grid (SRTM4 resampled to 100m) for about 3,700 viewpoints for an area of about 70,000 km² on a

standard computer. Even including earth curvature and atmospheric refraction is a matter of minutes. Most likely this is owed to the heuristic applied in QGIS, but it seems to be effective to increase raster resolution in the present study.

Thank you. Please see our responses to your comments above about the used resolution (25-m) of the elevation grid. For this high-resolution DEM data, we first carried out a demo application of the reverse-viewshed analysis only including 679 viewpoints. We did this using the `r.viewshed` function in GRASS-GIS. When earth curvature and atmospheric refraction are not considered the process took 24 minutes to finish on a standard PC (intel core i7). Calculation for 357,588 viewpoints with 25-m raster resolution would take over a week on conventional PCs. Furthermore, this does not account for the time to merge all the viewshed layers that meet the selected thresholds. During the demo application for the Aachen region alone, the merging process took 53 minutes for all visibility scenarios considered. For this reason, we parallelized the reverse viewshed analysis on our institute's cluster to speed up the process.

L 366: please explain the merging of R-ZTV maps: is it a logical OR function or do you use another logic?

Thank you. We used the logical OR function, where we combined with union operator all viewshed layers from viewpoints that meet the thresholds condition. We have now added an explanation to the sentence:

In line 474: *"The merge process combines all reverse viewshed raster files from viewpoints that meet the thresholds with a union operator at a resolution of 25 meter."*